# Training Deep Normalization-Free Spiking Neural Networks with Lateral Inhibition

**Peiyu Liu**[1,2], **Jianhao Ding**[1] **& Zhaofei Yu**[1,2,*]
[1] School of Computer Science, Peking University
[2] Institute for Artificial Intelligence, Peking University

## Abstract

Spiking Neural Networks (SNNs) have garnered significant attention as a central paradigm in neuromorphic computing, owing to their energy efficiency and biological plausibility. However, training deep SNNs has critically depended on explicit normalization schemes, leading to a trade-off between performance and biological realism. To resolve this conflict, we propose a normalization-free learning framework that incorporates lateral inhibition inspired by cortical circuits. Our framework replaces the traditional feedforward SNN layer with distinct excitatory (E) and inhibitory (I) neuronal populations that capture the key features of the cortical E-I interaction. The E-I circuit dynamically regulates neuronal activity through subtractive and divisive inhibition, which respectively control the excitability and gain of neurons. To stabilize end-to-end training of the biologically constrained SNNs, we propose two key techniques: E-I Init and E-I Prop. E-I Init is a dynamic parameter initialization scheme that balances excitatory and inhibitory inputs while performing gain control. E-I Prop decouples the backpropagation of the circuit from the forward pass, regulating gradient flow. Experiments across multiple datasets and network architectures demonstrate that our framework enables stable training of deep normalization-free SNNs with biological realism, achieving competitive performance. Therefore, our work not only provides a solution to training deep SNNs but also serves as a computational platform for further exploring the functions of E-I interaction in large-scale cortical computation. Code is available at `https://github.com/vwOvOwv/DeepEISNN`.

## 1 Introduction

Inspired by computational principles of biological neurons, Spiking Neural Networks (SNNs) stand at the intersection of artificial intelligence and neuroscience (Maass, 1997; Ghosh-Dastidar & Adeli, 2009). They not only enable highly energy-efficient computation on neuromorphic hardware (Roy et al., 2019; Xiao et al., 2025) but also provide models for understanding cortical computation across multiple scales (Kumarasinghe et al., 2021; Korcsak-Gorzo et al., 2022; N'dri et al., 2024). This duality places SNNs at the heart of the emerging field of NeuroAI (Sadeh & Clopath, 2025), fostering a synergy between artificial intelligence and neuroscience. While a deeper understanding of the brain's computational principles inspires novel SNN architectures (Fang et al., 2021b; Pan et al., 2025), advances in deep learning have concurrently enabled the training of large-scale, high-performance SNNs (Wu et al., 2018; Fang et al., 2021a; Bu et al., 2022). Some of these well-trained models, in turn, can serve as *in silico* platforms for investigating multi-scale cortical computation that is difficult to access through wet-lab experiments (Bellec et al., 2018).

Despite the promising synergy, realizing the potential of SNNs as an ideal platform for exploring both machine and biological intelligence is hindered by a trade-off between computational performance and biological plausibility. Many learning algorithms achieve high performance by adopting backpropagation-based techniques, treating spiking neurons as recurrent units unrolled over time (Neftci et al., 2019; Fang et al., 2021a). While this strategy yields models whose performance is comparable to their Artificial Neural Network (ANN) counterparts, it reduces SNNs to mere deep

---

*Corresponding author: yuzf12@pku.edu.cn

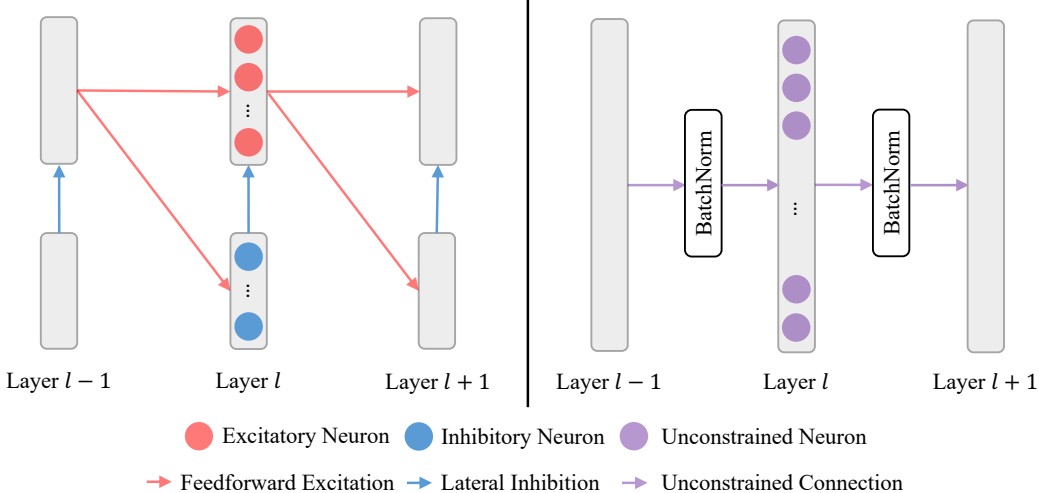

Figure 1: The proposed feedforward E-I circuit (**left**), compared with normalization-equipped architecture (**right**, BN as an example). Neurons in layer $l-1$ and $l+1$ are not shown.

learning artifacts, sacrificing basic biological properties. As a result, these models often ignore fundamental principles in neuroscience such as E-I dynamics, which are crucial in gain control (Goldwyn et al., 2018; Del Rosario et al., 2025), neural oscillation (Buzsáki & Draguhn, 2004), selective attention (Zhang et al., 2014), etc. Consequently, deriving meaningful insights for neuroscience from these biologically unfaithful models becomes a challenge.

However, approaches that prioritize biological realism also face the challenge of unstable training. Biologically plausible learning rules such as spike-timing-dependent plasticity (STDP) (Gerstner et al., 1996; Bi & Poo, 1998), often struggle with the instability during training and thus can only be applied to shallow SNNs (Habenschuss et al., 2012; Beyeler et al., 2013). In the field of deep learning, this training instability issue is partly overcome by explicit normalization schemes, most notably batch normalization (BN) (Ioffe & Szegedy, 2015). While such normalization schemes are powerful tools to accelerate and stabilize training, they explicitly collect statistics from inputs, which has no known biological analogue, making it implausible for brain-inspired models. This widens the gap between high performance and biological plausibility, highlighting the need for a biologically grounded alternative.

Here, we address the challenge of training deep biologically plausible SNNs by introducing lateral inhibition, a canonical interaction mechanism between excitatory and inhibitory neurons in cortex. We propose a normalization-free learning framework based on an E-I circuit, as shown in Figure 1. Our framework presents a brain-inspired alternative to standard normalization schemes, bridging the gap between high-performance deep learning and biologically plausible neural computation. The main contributions of our work are summarized as follows:

1. We incorporate a canonical E-I circuit, composed of distinct excitatory and inhibitory neuron populations, into deep SNNs to enable normalization-free training.

2. We introduce a dynamic initialization scheme to ensure effective learning from the very beginning of training.

3. We integrate adaptive stabilization of divisive inhibition and straight-through estimator (STE) into the framework. These techniques prove to be essential for stable learning of deep SNNs with the E-I circuit.

4. Experiments demonstrate that the framework achieves competitive performance across different datasets and architectures, indicating the viability of our brain-inspired learning algorithm.

## 2 RELATED WORK

### 2.1 NORMALIZATION IN SNNS

Recent methods for training deep SNNs can be broadly categorized into two approaches, ANN-to-SNN conversion (ANN2SNN) (Rueckauer et al., 2017; Sengupta et al., 2019; Han et al., 2020; Han & Roy, 2020; Ding et al., 2021; Stöckl & Maass, 2021; Bu et al., 2022; Zhao et al., 2025) and direct end-to-end training (Lee et al., 2016; Neftci et al., 2019; Li et al., 2021; Fang et al., 2021a;b; Guo et al., 2022b; Xiao et al., 2022; Zhu et al., 2024). Normalization is critical in both methods. Many ANN2SNN methods merge normalization parameters of ANNs into synaptic weights of spiking neurons (Sengupta et al., 2019; Han et al., 2020; Bu et al., 2022). In contrast, training SNNs from scratch usually directly adopts normalization schemes developed for ANNs, especially BN (Ioffe & Szegedy, 2015). There are also BN-derived normalization schemes designed for SNNs, like NeuNorm (Wu et al., 2019), BNTT (Kim & Panda, 2021), tdBN (Zheng et al., 2021), TEBN (Duan et al., 2022), and TAB (Jiang et al., 2024). However, these strategies still inherit the biological implausibility of normalization schemes due to their dependence on statistics collected from batch inputs throughout the training. Therefore, the need for fully brain-inspired normalization alternatives remains.

### 2.2 NEURAL NETWORKS WITH SEPARATE EXCITATORY AND INHIBITORY UNITS

The interaction between excitatory and inhibitory neurons has been a key topic in neuroscience (Haider et al., 2006; Ahmadian & Miller, 2021; Cohen Kadosh, 2025). Historically, computational models of these circuits have been confined to shallow networks, often focusing on the dynamics of a few interacting populations/neurons to explain basic principles (Somers et al., 1995; Wilson & Cowan, 1972; Carandini & Heeger, 2012). One of the reasons for this limitation is that training deep networks with the E-I circuit proves to be a significant challenge. This has left the whole picture of E-I dynamics largely unexplored. A remarkable step towards deep E-I networks was taken by Cornford et al. (2021). By developing techniques for parameter initialization, they demonstrated that ANNs with separate excitatory and inhibitory neuron populations could be effectively trained. However, the model is built fully with the rectified linear units (ReLU) and thus does not capture the temporal properties of the circuit. While SNNs can address temporal processing, such initialization techniques cannot be directly applied to SNNs with the E-I circuit.

## 3 PRELIMINARIES

### 3.1 EXCITATION AND INHIBITION IN CORTEX

A fundamental principle in the cortex is the functional segregation of neurons into distinct excitatory and inhibitory populations (Barranca et al., 2022). This circuit-level architecture implies that a given neuron typically exerts a uniform influence (either depolarization or hyperpolarization) on all its postsynaptic targets. When translating this principle to artificial neural networks, all outgoing synaptic weights from a given neuron should share the same sign. This makes standard initialization techniques like Xavier (Glorot & Bengio, 2010) and Kaiming (He et al., 2015) initialization inapplicable, as they sample weights from zero-centered distributions that assign both positive and negative weights to each neuron.

### 3.2 NEURON MODELS

**Excitatory neurons.** We model excitatory neurons with the widely adopted leaky-integrate-and-fire (LIF) model (Gerstner et al., 2014). For a given layer $l$ with $n_{\mathrm{E}}^{[l]}$ excitatory neurons (we use superscript $[l]$ to denote the $l$-th layer), the sub-threshold dynamics of the membrane potential $\mathbf{u}_{\mathrm{E}}^{[l]}(t) \in \mathbb{R}^{n_{\mathrm{E}}^{[l]}}$ are described by the following equation:

$$\tau_{\mathrm{E}} \frac{\mathrm{d}\mathbf{u}_{\mathrm{E}}^{[l]}(t)}{\mathrm{d}t} = -\left(\mathbf{u}_{\mathrm{E}}^{[l]}(t) - u_{\mathrm{E,rest}}\right) + \mathbf{I}_{\mathrm{E}}^{[l]}(t), \tag{1}$$

where $\tau_{\mathrm{E}}$ and $u_{\mathrm{E,rest}}$ are the membrane time constant and resting potential of all excitatory neurons, respectively. $\mathbf{I}_{\mathrm{E}}^{[l]}(t) \in \mathbb{R}^{n_{\mathrm{E}}^{[l]}}$ represents the input currents to the excitatory neurons at time $t$, assuming

a unit membrane resistance. For discrete-time simulation, we approximate Equation 1 using the first-order Euler method with a time step $\Delta t = 1$. By setting $u_{E,rest}$ to 0 and omitting decay of input currents, we obtain the following iterative update rule:

$$\mathbf{u}_E^{[l]}[t+1] = \left(1 - \frac{1}{\tau_E}\right)\mathbf{u}_E^{[l]}[t] + \mathbf{I}_E^{[l]}[t]. \tag{2}$$

An excitatory neuron emits a spike when its membrane potential exceeds a firing threshold $\theta_E$ (which is set to 1 for all excitatory neurons). To model the subsequent reset, we employ a soft reset mechanism where the potential of a firing neuron is reduced by $\theta_E$. This leads to full dynamics for excitatory neurons in layer $l$,

$$\mathbf{u}_E^{[l]}[t+1] = \left(1 - \frac{1}{\tau_E}\right)\left(\mathbf{u}_E^{[l]}[t] - \theta_E \cdot \mathbf{s}_E^{[l]}[t]\right) + \mathbf{I}_E^{[l]}[t], \tag{3}$$

where the spikes are generated by the Heaviside step function $H$, i.e., $\mathbf{s}_E^{[l]}[t] = H\left(\mathbf{u}_E^{[l]}[t] - \theta_E\right)$.

To distinguish this process from the dynamics of inhibitory neurons in layer $l$, we encapsulate it into an operator $\mathcal{F}_E^{[l]}$. This operator takes the input currents at the current time step as its arguments and produces the corresponding output spikes.

$$\mathbf{s}_E^{[l]}[t] = \mathcal{F}_E^{[l]}\left(\mathbf{I}_E^{[l]}[t]; \mathbf{u}_E^{[l]}[t], \tau_E, \theta_E\right). \tag{4}$$

**Inhibitory neurons.** Many inhibitory neurons, especially parvalbumin ($PV^+$) neurons, are known to be fast-spiking (FS), characterized by a much smaller membrane time constant $\tau_I$ compared to that of excitatory pyramidal neurons (Hu et al., 2014; Prince et al., 2021). In our discrete-time simulation, the time step $\Delta t = 1$ is chosen to be on a similar scale as the time constant of excitatory neurons (e.g., $\tau_E = 2$), which implies $\tau_I \ll \Delta t$ since $\tau_I \ll \tau_E$. Under this condition, the dynamics of inhibitory neurons can reach a steady state almost instantaneously within a single time step. This allows us to apply an approximation to LIF model by treating $\tau_I$ as negligible, which leads to

$$0 = -\left(\mathbf{u}_I^{[l]}[t] - u_{I,rest}\right) + \mathbf{I}_I^{[l]}[t]. \tag{5}$$

Here we use notations similar to those in the excitatory neuron model, and the subscript I denotes inhibitory neurons. By setting $u_{I,rest}$ to 0, we find the membrane potential of an inhibitory neuron is determined purely by its input currents at time $t$:

$$\mathbf{u}_I^{[l]}[t] = \mathbf{I}_I^{[l]}[t]. \tag{6}$$

Since this potential remains constant throughout duration $\Delta t$, the neurons can fire $\left\lfloor \max\left(0, \mathbf{I}_I^{[l]}[t]\right)/\theta_I \right\rfloor$ times if we apply soft reset and a fixed firing threshold $\theta_I$. Finally, by setting $\theta_I = 1$, we can directly model the total spike outputs of inhibitory neurons at time $t$ as

$$\mathbf{s}_I^{[l]}[t] = \left\lfloor \max\left(0, \mathbf{I}_I^{[l]}[t]\right) \right\rfloor \approx \max\left(0, \mathbf{I}_I^{[l]}[t]\right). \tag{7}$$

Similar to the excitatory neuron model, we encapsulate this process into an operator $\mathcal{F}_I^{[l]}$:

$$\mathbf{s}_I^{[l]}[t] = \mathcal{F}_I^{[l]}\left(\mathbf{I}_I^{[l]}[t]\right) \approx \max\left(0, \mathbf{I}_I^{[l]}[t]\right). \tag{8}$$

See Appendix D.1 for a detailed derivation.

## 4 METHODS

### 4.1 E-I CIRCUIT IN SNNS

The model is constructed according to the canonical E-I circuit shown in Figure 1. Each layer $l$ comprises $n_E^{[l]}$ excitatory neurons and $n_I^{[l]}$ inhibitory neurons, where $n_E^{[l]}/n_I^{[l]} = 4$ according to biological evidence (Markram et al., 2004). Computation performed by this circuit at each time step $t$ is demonstrated in Figure 2.

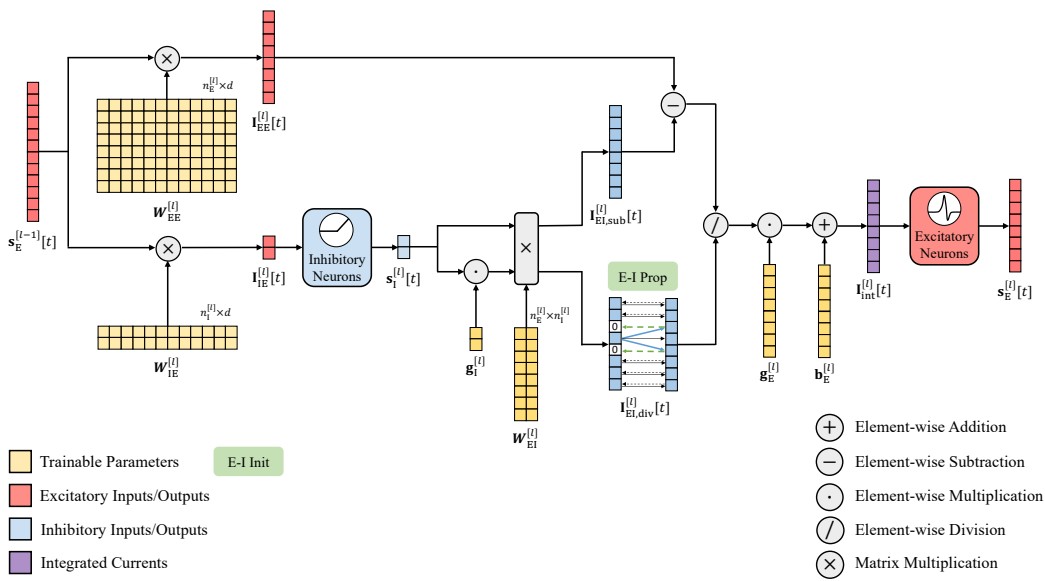

Figure 2: An overview of the proposed framework. E-I Init enables effective learning from the very beginning through a dynamic parameter initialization scheme. E-I Prop then ensures stable end-to-end training by regulating the forward and backward passes. For the sake of brevity, our analysis in the main text focuses on a fully connected architecture. Extension of our method to convolutional neural networks (CNNs) is detailed in Appendix E.1.

**Excitatory projections.** First, excitatory population in layer $l - 1$ undergoes dynamics described by Equation 1 to Equation 4 and emits spikes $\mathbf{s}_{\mathrm{E}}^{[l-1]}[t] \in \{0,1\}^d$, where $d$ is the dimension of input spikes. These input spikes induce two excitatory currents into both the excitatory and inhibitory populations of layer $l$:

$$\mathbf{I}_{\mathrm{EE}}^{[l]}[t] = \boldsymbol{W}_{\mathrm{EE}}^{[l]}\mathbf{s}_{\mathrm{E}}^{[l-1]}[t], \tag{9}$$

$$\mathbf{I}_{\mathrm{IE}}^{[l]}[t] = \boldsymbol{W}_{\mathrm{IE}}^{[l]}\mathbf{s}_{\mathrm{E}}^{[l-1]}[t]. \tag{10}$$

Here, subscript AB denotes projections from population B to population A. $\boldsymbol{W}_{\mathrm{EE}}^{[l]} \in \mathbb{R}^{n_{\mathrm{E}}^{[l]} \times d}$ and $\boldsymbol{W}_{\mathrm{IE}}^{[l]} \in \mathbb{R}^{n_{\mathrm{I}}^{[l]} \times d}$ are corresponding synaptic weight matrices, which are constrained to be non-negative during training due to E-I segregation.

**Lateral inhibition.** Following the fast-spiking approximation in Section 3.2, the activity of inhibitory neurons is modeled by $\mathcal{F}_{\mathrm{I}}^{[l]}$,

$$\mathbf{s}_{\mathrm{I}}^{[l]}[t] = \mathcal{F}_{\mathrm{I}}^{[l]}\left(\mathbf{I}_{\mathrm{IE}}^{[l]}[t]\right). \tag{11}$$

This inhibitory signal then laterally regulates the excitatory population. Motivated by the biophysical distinction between dendritic and somatic inhibition, we decompose this regulation into subtractive inhibition for E-I balance and divisive inhibition for gain control.

$$\mathbf{I}_{\mathrm{EI,sub}}^{[l]}[t] = \boldsymbol{W}_{\mathrm{EI}}^{[l]}\mathbf{s}_{\mathrm{I}}^{[l]}[t], \tag{12}$$

$$\mathbf{I}_{\mathrm{EI,div}}^{[l]}[t] = \boldsymbol{W}_{\mathrm{EI}}^{[l]}\left(\mathbf{g}_{\mathrm{I}}^{[l]} \odot \mathbf{s}_{\mathrm{I}}^{[l]}[t]\right), \tag{13}$$

where $\boldsymbol{W}_{\mathrm{EI}}^{[l]} \in \mathbb{R}^{n_{\mathrm{E}}^{[l]} \times n_{\mathrm{I}}^{[l]}}$ is the weight matrix for inhibitory-to-excitatory projections, $\mathbf{g}_{\mathrm{I}}^{[l]} \in \mathbb{R}^{n_{\mathrm{I}}^{[l]}}$ is a trainable parameter modulating the strength of divisive inhibition, and $\odot$ denotes the Hadamard product.

**Input integration and spiking.** Finally, the excitatory population integrates the excitatory currents with both forms of inhibition to compute the total input currents.

$$\mathbf{I}_{\mathrm{int}}^{[l]}[t] = \mathbf{g}_{\mathrm{E}}^{[l]} \odot \frac{\mathbf{I}_{\mathrm{EE}}^{[l]}[t] - \mathbf{I}_{\mathrm{EI,sub}}^{[l]}[t]}{\mathbf{I}_{\mathrm{EI,div}}^{[l]}[t]} + \mathbf{b}_{\mathrm{E}}^{[l]}, \tag{14}$$

where $\mathbf{g}_{\mathrm{E}}^{[l]}, \mathbf{b}_{\mathrm{E}}^{[l]} \in \mathbb{R}^{n_{\mathrm{E}}^{[l]}}$ are trainable parameters, and the division is performed element-wise. Taking the integrated currents as input currents, excitatory neurons emit spikes to the next layer,

$$\mathbf{s}_{\mathrm{E}}^{[l]}[t] = \mathcal{F}_{\mathrm{E}}^{[l]} \left( \mathbf{I}_{\mathrm{int}}^{[l]}[t]; \mathbf{u}_{\mathrm{E}}^{[l]}[t], \tau_{\mathrm{E}}, \theta_{\mathrm{E}} \right). \tag{15}$$

## 4.2 E-I INIT: DYNAMIC PARAMETER INITIALIZATION

As discussed in Section 3.1, standard zero-centered initialization schemes like Xavier (Glorot & Bengio, 2010) and Kaiming (He et al., 2015) are incompatible with the strict sign constraints of E-I segregation. Some naive initializations under this constraint lead to pathological network activity, making the training of deep architectures infeasible (see Section 5.2). Therefore, we propose E-I Init, an initialization scheme designed for deep SNNs with the E-I circuit. Its design is guided by two primary objectives: (1) establishing an initial E-I balance to prevent neurons from silencing or saturating, and (2) setting an appropriate initial gain to ensure stable signal propagation.

**E-I balance via subtractive inhibition.** A key goal of our initialization scheme is to ensure that neurons operate in a responsive regime. We achieve this by setting the expected subtractive inhibitory currents to approximately balance the expected excitatory currents, which is defined as

$$\mathbb{E}\left[\mathbf{I}_{\mathrm{EE},i}^{[l]}\right] \approx \mathbb{E}\left[\mathbf{I}_{\mathrm{EI,sub},i}^{[l]}\right], \tag{16}$$

for each excitatory neuron $i$ in layer $l$ and results in a near-zero expected net input, preventing neurons from being saturated or silent at initialization. To implement this under the constraint of E-I segregation, we draw inspiration from Cornford et al. (2021) and leverage the exponential distribution for weight initialization. Specifically, we draw the excitatory weights $\mathbf{W}_{\mathrm{EE}}^{[l]}$ and $\mathbf{W}_{\mathrm{IE}}^{[l]}$ from an exponential distribution with rate parameter $\lambda^{[l]}$. The inhibitory weights $\mathbf{W}_{\mathrm{EI}}^{[l]}$ are deterministically set to $1/n_{\mathrm{I}}^{[l]}$ to uniformly distribute the inhibitory signals. Here $\mathbf{W}_{\mathrm{AB}}^{[l]} \in \mathbb{R}$ denotes elements of $\boldsymbol{W}_{\mathrm{AB}}^{[l]}$. Assuming that presynaptic neurons fire independently with an average probability of $p$ (i.e., the spike from each neuron at any time step is an independent and identically distributed (i.i.d.) Bernoulli trial with probability $p$), the expected excitatory input to excitatory neuron $i$ in layer $l$ is

$$\mathbb{E}\left[\mathbf{I}_{\mathrm{EE},i}^{[l]}\right] = dp\mathbb{E}\left[\mathbf{W}_{\mathrm{EE}}^{[l]}\right], \tag{17}$$

where $d$ is the dimension of input spikes. Similarly, the expected subtractive inhibitory currents are

$$\mathbb{E}\left[\mathbf{I}_{\mathrm{EI,sub},i}^{[l]}\right] = n_{\mathrm{I}}^{[l]} dp\mathbb{E}\left[\mathbf{W}_{\mathrm{IE}}^{[l]}\right] \mathbb{E}\left[\mathbf{W}_{\mathrm{EI}}^{[l]}\right]. \tag{18}$$

Therefore, by setting $\mathbb{E}\left[\mathbf{W}_{\mathrm{EE}}^{[l]}\right] = \mathbb{E}\left[\mathbf{W}_{\mathrm{IE}}^{[l]}\right] = 1/\lambda^{[l]}$ and $\mathbf{W}_{\mathrm{EI}}^{[l]} = 1/n_{\mathrm{I}}^{[l]}$, we arrive at the desired balance defined by Equation 16 (see Appendix D.2 for a detailed derivation).

**Gain control via divisive inhibition.** Our second objective is establishing stable signal propagation by setting an appropriate initial gain for excitatory neurons. In our model, gain is primarily modulated by the divisive inhibitory currents. Inspired by normalization techniques, our strategy is modulating divisive inhibition through $\mathbf{g}_{\mathrm{I}}^{[l]}$ such that the expected value of this divisive inhibitory currents approximates the standard deviation of the excitatory inputs. This can be formulated as

$$\mathbb{E}\left[\mathbf{I}_{\mathrm{EI,div},i}^{[l]}\right] = \mathrm{std}\left(\mathbf{I}_{\mathrm{EE},i}^{[l]}\right) \tag{19}$$

for each excitatory neuron $i$ in layer $l$. In this way, the divisive operation effectively scales the final input, leading the excitatory neurons to a responsive regime at initialization. Similar to Equation 18, for inhibitory neuron $i$ in layer $l$,

$$\mathbb{E}\left[\mathbf{I}_{\mathrm{EI,div},i}^{[l]}\right] = n_{\mathrm{I}}^{[l]} dp\mathbb{E}\left[\mathbf{g}_{\mathrm{I}}^{[l]}\right] \mathbb{E}\left[\mathbf{W}_{\mathrm{IE}}^{[l]}\right] \mathbb{E}\left[\mathbf{W}_{\mathrm{EI}}^{[l]}\right] = \frac{dp\mathbb{E}\left[\mathbf{g}_{\mathrm{I}}^{[l]}\right]}{\lambda^{[l]}}. \tag{20}$$

By assuming that the spike from each neuron at any time step is i.i.d. Bernoulli trial with probability $p$, the standard deviation of excitatory input currents of neuron $i$ can be formulated as

$$\text{std}\left(\mathbf{I}_{\text{EE},i}^{[l]}\right) = \frac{\sqrt{dp(2-p)}}{\lambda^{[l]}}, \tag{21}$$

see Appendix D.3 for details. Therefore, by setting each element of $\mathbf{g}_{\text{I}}^{[l]}$ to $\sqrt{\frac{2-p}{dp}}$, we achieve a normalization effect at initialization, making the effective training of deep SNNs possible.

**Initialization of other parameters.** Finally, we initialize $\lambda^{[l]} = \sqrt{\frac{d(2-p)}{1-p}}$ by setting the condition $\text{std}\left(\mathbf{I}_{\text{EE},i}^{[l]}\right) = \sqrt{p(1-p)}$ (see Appendix D.4). $\mathbf{g}_{\text{E}}^{[l]}$ and $\mathbf{b}_{\text{E}}^{[l]}$ are initialized as vector $\mathbf{1}$ and $\mathbf{0}$, respectively.

**Dynamic firing probability estimation**. Since the initialization depends on the averaged firing probability $p$, we use the first batch in training set to compute point estimations of $p$ and other statistics, leading to a dynamic initialization regime (see Appendix E.2 for implementation details).

### 4.3 E-I PROP: STABILIZING END-TO-END TRAINING

While E-I Init provides a stable initial state, the interplay between divisive inhibition and discrete input spikes induces training instabilities. These instabilities arise mainly from two sources: first, near-zero divisive currents in the forward pass trigger numerical explosions; and second, disproportionately large gradients in the lateral inhibitory pathway destabilize training. To overcome these issues, we propose E-I Prop, a toolkit that decouples the backpropagation of the E-I circuit from the forward pass, regulating gradient flow. The forward stability is ensured by adaptive divisive inhibition, while the backward stability is achieved through a straight-through estimator (STE) combined with gradient scaling.

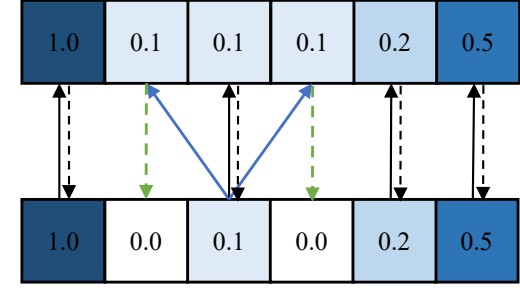

Figure 3: Mechanism of adaptive stabilization and STE. **Forward (bottom-up):** The adaptive stabilization handles numerical instability by dynamically replacing zero elements in the denominator with the smallest positive value in the sample, preserving maximal dynamic range. **Backward (top-down):** The STE allows gradients to bypass the replacement operation, treating it as an identity function.

**Adaptive stabilization of divisive inhibition.** To prevent division-by-zero error in the forward pass, a common technique is adding a small constant $\epsilon$ to the denominator. However, a fixed $\epsilon$ is ill-suited for our network because the divisive inhibitory currents are designed to provide a suitable dynamic range and perform gain control. A pre-defined $\epsilon$ that is too small may fail to prevent numerical instability if the denominator collapses towards zero, while one that is too large will artificially suppress the dynamic range by dominating the denominator. As shown in Figure 3, here we propose an adaptive stabilization method. Instead of using a static constant, our approach sets a dynamic, input-dependent lower bound for the denominator. Specifically, for each sample within a batch, we identify any zero values in $\mathbf{I}_{\text{EI,div}}$. Then, these zero values are replaced by the smallest positive value found within the same sample, which proves to be necessary for effective training (see Section 5.2 for details).

**Straight-through estimator (STE).** Since the replacement operation in our adaptive stabilization is non-differentiable, it misdirects the gradient flow and destabilizes learning. To address this, we employ STE, a common technique for handling non-differentiable operations in neural networks (Bengio et al., 2013). In the forward pass, we perform the adaptive stabilization as described above. In the backward pass, we approximate the derivative of this non-differentiable operation with an identity function (see Appendix E.3 for implementation). This approach decouples the forward-pass requirement for numerical stability from the backward-pass requirement for a clean gradient path, ensuring that the network can learn robustly.

**Gradient scaling.** To ensure stable training, it is crucial to balance the influence of the feedforward excitatory and lateral inhibitory pathways on parameter updates. Both theoretical and empirical analyses reveal that gradients for the lateral inhibitory weight, $W_{\mathrm{EI}}^{[l]}$, are disproportionately larger than those for other synaptic weights (see Appendix D.5 and F.1). To counteract this gradient amplification, we scale the gradients of $W_{\mathrm{EI}}^{[l]}$ by a factor of $1/d$, where $d$ is the dimension of the input spikes. This effectively balances the update magnitudes between the two pathways.

As illustrated in Figure 2, our method provides a completely normalization-free learning framework composed of E-I Init and E-I Prop, enabling stable and effective end-to-end training from scratch.

## 5 EXPERIMENTS

### 5.1 PERFORMANCE ON CLASSIFICATION TASKS

We evaluate our framework on both static and event-based datasets, with results summarized in Table 1. Our ResNet-18 model (He et al., 2016; Fang et al., 2021a) achieves 92.05±0.11% top-1 accuracy on CIFAR-10 (Krizhevsky, 2009), surpassing all normalization-free baselines. Notably, on the more challenging TinyImageNet (Le & Yang, 2015), the model attains 50.29% accuracy, demonstrating scalability to large-scale datasets. Furthermore, to validate temporal processing capabilities, we test on neuromorphic datasets where our method achieves 94.86±0.86% on DVS-Gesture (Amir et al., 2017) and 77.66±0.48% on CIFAR10-DVS (Li et al., 2017), outperforming several BN-based methods. Collectively, these results across diverse datasets confirm that our E-I circuit, empowered by E-I Init and E-I Prop, serves as a robust and scalable alternative to explicit normalization in deep SNNs.

### 5.2 ABLATION STUDY

Results of ablation study are summarized in Table 2. It confirms that each component in our method is indispensable for stable and high-performance training. Replacing E-I constraint (on sign of weights) or E-I Init by applying a standard/clamped Kaiming initialization (He et al., 2015) leads to either a complete training failure or a significant accuracy drop, proving its necessity for establishing a proper E-I balance and gain control through E-I segregation and E-I init. Furthermore, the common $\epsilon$-stabilization fails across all tested values, confirming that our adaptive stabilization mechanism is crucial for maintaining both numerical stability and gain control. Finally, removing gradient scaling on $W_{\mathrm{EI}}$ causes training collapse, which validates its role in stabilizing learning dynamics. These results demonstrate that E-I constraint, E-I Init, and E-I Prop for effective training of deep E-I SNNs.

### 5.3 BIMODAL REPRESENTATION LEARNED BY THE E-I CIRCUIT

Visualization of the integrated currents distribution reveals that our framework leverages a normalization-like effect at initialization but ultimately learns a more sophisticated representation. As shown in Figure 4, E-I Init successfully produces stable, zero-centered Gaussian-like distributions at the beginning of training. However, after training, some of the distributions evolve into a distinct bimodal shape, in contrast to the Gaussian-like outputs of SNNs with vanilla BN (see Appendix F.3).

This emergent bimodality can be considered as a mixture of two Gaussian-like distributions, with one distribution centered at the negative regime. This indicates that the E-I circuit is not equivalent to simple normalization, but rather a dynamic separation of neurons into activated and suppressed populations, confirming its role as a strategy distinct from standard normalization.

Table 1: Comparison with other E-I constrained, normalization-free, and BN-equipped methods.

| Dataset | E-I | Normalization | Method | Architecture | Time steps | Top-1 Accuracy (%) |
|---------|-----|---------------|--------|--------------|------------|--------------------|
| **CIFAR-10** | × | × | DAP (Micheli et al., 2025) | 7-layer CNN | 3 | 57.52 |
| | | | SRI (Ding et al., 2025) | VGG-9 | 20 | 87.62 |
| | | | B-SNN (Karimah et al., 2025) | VGG-8 | 64 | 87.73 |
| | | | EICIL (Shao et al., 2023) | ResNet-18 | – | 90.34 |
| | | | IM-Loss (Guo et al., 2022a) | CIFARNet | 4 | 90.90 |
| | × | ✓ | *Vanilla BN | VGG-16 | 4 | 94.29 ± 0.07 |
| | | | | VGG-19 | 4 | 94.11 ± 0.15 |
| | | | | ResNet-18 | 4 | 95.37 ± 0.13 |
| | | | BNTT (Kim & Panda, 2021) | VGG-9 | 20 | 90.30 |
| | | | NeuNorm (Wu et al., 2019) | CIFARNet | 12 | 90.53 |
| | | | TEBN (Duan et al., 2022) | VGG-9 | 4 | 92.81 |
| | | | | ResNet-19 | 4 | 94.70 |
| | | | tdBN (Zheng et al., 2021) | ResNet-19 | 4 | 92.92 |
| | | | TAB (Jiang et al., 2024) | VGG-9 | 4 | 93.41 |
| | | | | ResNet-19 | 4 | 94.76 |
| | ✓ | × | FDI (Rossbroich et al., 2022) | 6-layer CNN | 50 | 65.60 |
| | | | *DANN (Cornford et al., 2021) | VGG-16 | – | 88.54 ± 0.38 |
| | | | | VGG-19 | – | 88.28 ± 0.27 |
| | | | EICIL (Shao et al., 2023) | E-I Net | – | 89.43 |
| | | | BackEISNN (Zhao et al., 2022) | 5-layer CNN | 20 | 90.93 |
| | | | **DeepEISNN (Ours)** | **VGG-16** | **4** | **90.80 ± 0.16** |
| | | | | **VGG-19** | **4** | **90.93 ± 0.33** |
| | | | | **ResNet-18** | **4** | **92.05 ± 0.11** |
| **CIFAR-100** | × | × | SRI (Ding et al., 2025) | VGG-11 | 20 | 54.94 |
| | | | EICIL (Shao et al., 2023) | ResNet-18 | – | 63.47 |
| | × | ✓ | BNTT (Kim & Panda, 2021) | VGG-11 | 50 | 66.60 |
| | | | TEBN (Duan et al., 2022) | VGG-11 | 4 | 74.37 |
| | | | TAB (Jiang et al., 2024) | VGG-11 | 4 | 75.89 |
| | ✓ | × | EICIL (Shao et al., 2023) | E-I Net | – | 53.86 |
| | | | **DeepEISNN (Ours)** | **VGG-16** | **4** | **64.95 ± 1.14** |
| | | | | **VGG-19** | **4** | **64.31 ± 0.41** |
| **CIFAR10-DVS** | × | × | IM-Loss (Guo et al., 2022a) | ResNet-19 | 10 | 72.60 |
| | × | ✓ | NeuNorm (Wu et al., 2019) | 7-layer CNN | 40 | 60.50 |
| | | | BNTT (Kim & Panda, 2021) | 7-layer CNN | 20 | 63.20 |
| | | | tdBN (Zheng et al., 2021) | ResNet-19 | 10 | 67.80 |
| | | | TEBN (Duan et al., 2022) | 7-layer CNN | 10 | 75.10 |
| | | | TAB (Jiang et al., 2024) | 7-layer CNN | 4 | 76.70 |
| | ✓ | × | **DeepEISNN (Ours)** | **VGG-8** | **10** | **77.30 ± 0.64** |
| | | | | **VGG-11** | **10** | **77.66 ± 0.48** |
| **DVS-Gesture** | ✓ | × | FDI (Rossbroich et al., 2022) | 6-layer CNN | 500 | 86.70 |
| | | | **DeepEISNN (Ours)** | **VGG-8** | **16** | **94.86 ± 0.86** |
| **TinyImageNet** | ✓ | × | **DeepEISNN (Ours)** | **ResNet-18** | **4** | **50.29** |

* Results marked with an asterisk are reproduced (averaged over 5 independent runs). Other results are cited from existing literature.

Table 2: Ablation study on individual components of the method on CIFAR-10 with VGG-8.

| Ablation Setting | | | | | |
|---|---|---|---|---|---|
| E-I Cons. | E-I Init | Adap. Stab. | Grad. Scale | Ablation Details | Top-1 Accuracy (%) |
| ✓ | ✓ | ✓ | ✓ | **Proposed Method (Ours)** | **87.03** |
| ✓ | × | ✓ | ✓ | Kaiming Init on $\boldsymbol{W}_{\mathrm{EE}}$ and $\boldsymbol{W}_{\mathrm{IE}}$ (clamped to $[0, +\infty)$) | 85.61 |
| ✓ | ✓ | × | ✓ | Fixed $\epsilon = 10^{-8}$ | Failed to converge |
| | | | | Fixed $\epsilon = 10^{-7}$ | Failed to converge |
| | | | | Fixed $\epsilon = 10^{-6}$ | Collapsed (Epoch 22) |
| | | | | Fixed $\epsilon = 10^{-5}$ | Collapsed (Epoch 5) |
| ✓ | ✓ | ✓ | × | No scaling on $\boldsymbol{W}_{\mathrm{EI}}$ | Collapsed (Epoch 10) |
| × | × | ✓ | ✓ | Kaiming Init on $\boldsymbol{W}_{\mathrm{EE}}$ and $\boldsymbol{W}_{\mathrm{IE}}$ (no sign constraint) | Failed to converge |

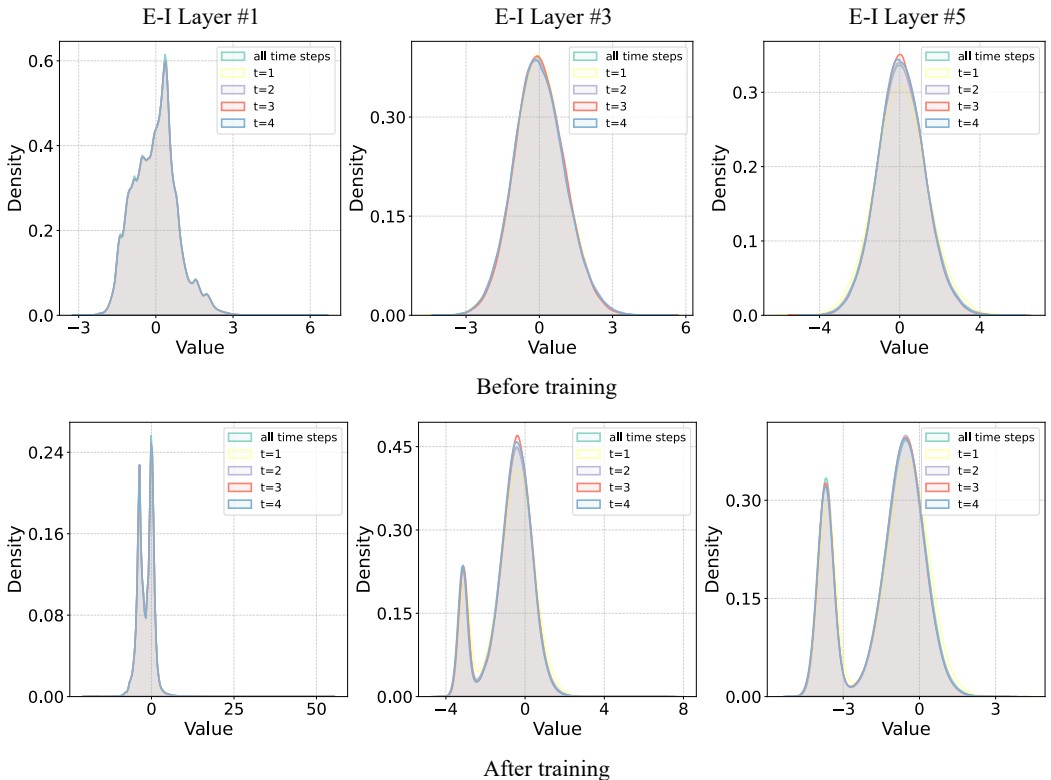

Figure 4: Distributions of the integrated input currents in the first, third and fifth layers of our model before and after training.

## 6 CONCLUSION AND DISCUSSION

In this work, we address the critical challenge in training deep normalization-free SNNs by introducing biologically inspired E-I segregation and lateral inhibition. Through a fine-grained initialization scheme E-I Init, and a toolkit of stabilization techniques E-I Prop, we enable the stable end-to-end training of deep normalization-free SNNs that capture features of canonical E-I circuit in the cortex. Our experiments demonstrate that the framework not only achieves competitive performance on multiple datasets but also learns a sophisticated mechanism of activity regulation that is functionally distinct from standard normalization. We show that the E-I circuit starts at an initial normalization-like state, and ultimately learns an activity regulation mechanism distinct from explicit normalizations like BN. Therefore, our work provides both a practical solution for building powerful, normalization-free SNNs and a compelling computational model for exploring how canonical cortical circuits perform complex and large-scale computation, further bridging the gap between deep learning and neuroscience.

While our framework successfully eliminates the need for normalization during training and inference, the deployment of the E-I circuit on digital neuromorphic hardware requires approximation techniques, such as bit-shifts or look-up tables. While there are obstacles for digital chips, emerging analog or mixed-signal platforms like DYNAP-SE2 (Richter et al., 2024) have already supported such operations. Therefore, our framework offers direct algorithmic compatibility with next-generation analog or mixed-signal neuromorphic hardware.

ACKNOWLEDGMENTS

This work is supported by STI 2030-Major Projects 2025ZD0217101, the National Natural Science Foundation of China (62422601, U24B20140), Beijing Municipal Science and Technology Program (Z241100004224004), Beijing Nova Program (20230484362, 20240484703), National Key Laboratory for Multimedia Information Processing, and Beijing Key Laboratory of Brain-inspired Spiking Large Models.

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

## A    LLM USAGE

We utilize LLMs to assist in proofreading and enhancing the clarity and readability of the manuscript. The core scientific contributions, experimental design, and data analysis are conducted entirely by the authors.

## B    REPRODUCIBILITY STATEMENT

All essential details regarding datasets, model architectures, and training hyperparameters are provided in Section 5 and Appendix F.6 to ensure reproducibility. The source code for our framework has been made publicly available at `https://github.com/vwOvOwv/DeepEISNN`.

## C    ETHICS STATEMENT

This work is foundational research focused on learning algorithms for brain-inspired neural networks and exclusively uses standard public datasets. We do not foresee any direct negative societal impacts or ethical concerns arising from this research.

## D    DETAILED MATHEMATICAL DERIVATIONS

This section provides detailed derivations for mathematical formulations introduced in the main text, using same notations.

### D.1    DYNAMICS OF THE FAST-SPIKING INHIBITORY NEURONS

In this section, we provide a formal derivation justifying the modelling of inhibitory neurons as stateless ReLU-like units in our discrete-time simulation. We demonstrate that this approximation is mathematically rigorous under the condition $\tau_I \ll \Delta t$. The sub-threshold dynamics of a LIF inhibitory neuron are governed by the following differential equation:

$$\tau_I \frac{d\mathbf{u}_I^{[l]}(t)}{dt} = -\left(\mathbf{u}_I^{[l]}(t) - u_{I,\text{rest}}\right) + \mathbf{I}_I^{[l]}(t), \tag{22}$$

where $\tau_I$ is the membrane time constant, $u_{I,\text{rest}}$ is the resting potential, and $\mathbf{I}_I^{[l]}(t)$ is the input currents. Assuming that $u_{I,\text{rest}} = 0$ and the input currents remain constant at $\mathbf{I}_0$ over the duration of a discrete time step $\Delta t$ (starting from time $t$), the analytical solution for the membrane potential at time $t + \Delta t$ is

$$\mathbf{u}_I^{[l]}(t + \Delta t) = \mathbf{u}_I^{[l]}(t)e^{-\frac{\Delta t}{\tau_I}} + \mathbf{I}_0\left(1 - e^{-\frac{\Delta t}{\tau_I}}\right). \tag{23}$$

In our simulation setup, excitatory neurons typically have a time constant $\tau_E = 2$, which is on the same order of magnitude as the simulation time step $\Delta t = 1$. In contrast, inhibitory neurons in our model represent biological fast-spiking interneurons (e.g., PV+ neurons), which are characterized by significantly smaller time constants compared to excitatory pyramidal neurons (Hu et al., 2014; Prince et al., 2021). This biological property implies the condition $\tau_I \ll \tau_E$, and consequently, $\tau_I \ll \Delta t$. Under this condition, the ratio $\frac{\Delta t}{\tau_I}$ becomes very large, causing the exponential decay factor to approach zero:

$$\lim_{\tau_I \to 0} e^{-\frac{\Delta t}{\tau_I}} = 0. \tag{24}$$

Substituting this limit into the update rule, the term $\mathbf{u}_I^{[l]}(t)$ vanishes, and the equation simplifies to:

$$\mathbf{u}_I^{[l]}(t + 1) \approx \mathbf{I}_0. \tag{25}$$

This result indicates that the membrane potential reaches a steady state determined entirely by the input currents almost instantaneously within a single time step. Consequently, the inhibitory neurons effectively converge to stateless units that do not carry temporal information across time steps.

Furthermore, the simulation time step $\Delta t$ is much longer than the intrinsic dynamics of the inhibitory neurons, which allows them to fire multiple times within a single step when the input is strong. By

applying a firing threshold of 1 and soft-reset mechanism, the total spike count at time $t+1$ is given by

$$\mathbf{s}_{\mathrm{I}}^{[l]}(t+1) = \left\lfloor \max\left(0, \mathbf{u}_{\mathrm{I}}^{[l]}(t+1)\right) \right\rfloor \approx \lfloor \max\left(0, \mathbf{I}_0\right) \rfloor \approx \max\left(0, \mathbf{I}_0\right). \qquad (26)$$

This derivation justifies the ReLU-like approximation for inhibitory neurons in the main text. Importantly, since the transient response is completed within a single step $\Delta t$, there is no accumulation of approximation error over time. The validity of this model is strictly determined by the timescale separation $\tau_{\mathrm{I}} \ll \Delta t$, making it a reasonable approximation.

## D.2 Initialization of $\boldsymbol{W}_{\mathrm{EE}}^{[l]}, \boldsymbol{W}_{\mathrm{IE}}^{[l]}, \boldsymbol{W}_{\mathrm{EI}}^{[l]}$

Our primary goal is to achieve a zero-mean expected net input for each excitatory neuron $i$ in layer $l$ (Equation 16) at initialization. From Equation 9 and Equation 10 we have

$$\mathbf{I}_{\mathrm{EE},i}^{[l]}[t] = \sum_{j=1}^{d} \mathrm{W}_{\mathrm{EE},ij}^{[l]} \mathbf{s}_{\mathrm{E},j}^{[l-1]}[t], i = 1, 2, \ldots, n_{\mathrm{E}}^{[l]}, \qquad (27)$$

$$\mathbf{I}_{\mathrm{IE},i}^{[l]}[t] = \sum_{j=1}^{d} \mathrm{W}_{\mathrm{IE},ij}^{[l]} \mathbf{s}_{\mathrm{E},j}^{[l-1]}[t], i = 1, 2, \ldots, n_{\mathrm{I}}^{[l]}. \qquad (28)$$

Here $\mathrm{W}_{\mathrm{EE},ij}^{[l]}$ and $\mathrm{W}_{\mathrm{IE},ij}^{[l]}$ denote the $(i,j)$ element of $\boldsymbol{W}_{\mathrm{EE}}^{[l]}$ and $\boldsymbol{W}_{\mathrm{IE}}^{[l]}$, respectively. Other notations are consistent with those in the main text. Then similarly, from Equation 11 and Equation 12 we obtain

$$\mathbf{I}_{\mathrm{EI,sub},i}^{[l]}[t] = \sum_{j=1}^{n_{\mathrm{I}}^{[l]}} \mathrm{W}_{\mathrm{EI},ij}^{[l]} \mathbf{s}_{\mathrm{I},j}^{[l]}[t] \approx \sum_{j=1}^{n_{\mathrm{I}}^{[l]}} \mathrm{W}_{\mathrm{EI},ij}^{[l]} \mathbf{I}_{\mathrm{IE},j}^{[l]}[t] \qquad (29)$$

$$= \sum_{j=1}^{n_{\mathrm{I}}^{[l]}} \mathrm{W}_{\mathrm{EI},ij}^{[l]} \sum_{k=1}^{d} \mathrm{W}_{\mathrm{IE},jk}^{[l]} \mathbf{s}_{\mathrm{E},k}^{[l-1]}[t], i = 1, 2, \ldots, n_{\mathrm{E}}^{[l]}. \qquad (30)$$

Note that $\mathbf{s}_{\mathrm{I},j}^{[l]}[t] = \mathcal{F}_{\mathrm{I}}^{[l]}\left(\mathbf{I}_{\mathrm{IE},j}^{[l]}[t]\right) \approx \max\left(0, \mathbf{I}_{\mathrm{IE},j}^{[l]}[t]\right) = \mathbf{I}_{\mathrm{IE},j}^{[l]}[t]$ since the elements of $\boldsymbol{W}_{\mathrm{IE}}^{[l]}$ and $\mathbf{s}_{\mathrm{E}}^{[l-1]}$ are non-negative. Therefore, by assuming $\mathbf{s}_{\mathrm{E},k}^{[l-1]}[t] \overset{\mathrm{i.i.d.}}{\sim} \mathrm{Bern}(p)$ and elements of weights are i.i.d. at initialization, we obtain Equation 17 and Equation 18.

## D.3 Initialization of $\mathbf{g}_{\mathrm{I}}^{[l]}$

$\mathbf{g}_{\mathrm{I}}^{[l]}$ is a gain factor modulating the strength of divisive inhibition. As mentioned in the main text, we initialize it by setting the condition $\mathbb{E}\left[\mathbf{I}_{\mathrm{EI,div},i}^{[l]}\right] = \mathrm{std}\left(\mathbf{I}_{\mathrm{EE},i}^{[l]}\right)$.

Similar to the derivation of $\mathbf{I}_{\mathrm{EI,sub},i}^{[t]}[t]$ in Appendix D.2,

$$\mathbb{E}\left[\mathbf{I}_{\mathrm{EI,div},i}^{[l]}\right] = n_{\mathrm{I}}^{[l]} dp \mathbb{E}\left[\mathbf{g}_{\mathrm{I}}^{[l]}\right] \mathbb{E}\left[\mathrm{W}_{\mathrm{IE}}^{[l]}\right] \mathbb{E}\left[\mathrm{W}_{\mathrm{EI}}^{[l]}\right] = \frac{dp \mathbb{E}\left[\mathbf{g}_{\mathrm{I}}^{[l]}\right]}{\lambda^{[l]}}. \qquad (31)$$

Below we derive the standard deviation of $\mathbf{I}_{\mathrm{EE},i}^{[l]}$. The variance of the excitatory input currents to neuron $i$ is

$$\mathrm{Var}\left(\mathbf{I}_{\mathrm{EE},i}^{[l]}\right) = \mathrm{Var}\left(\sum_{j=1}^{d} \mathrm{W}_{\mathrm{EE},ij}^{[l]} s_{\mathrm{E},j}^{[l-1]}\right), \qquad (32)$$

where $\mathrm{W}_{\mathrm{EE},ij}^{[l]}$ denotes the $(i,j)$ element of $\boldsymbol{W}_{\mathrm{EE}}^{[l]}$. Assuming the terms $\mathrm{W}_{\mathrm{EE},ij}^{[l]} s_{\mathrm{E},j}^{[l-1]}$ are independent for each $j$, the variance of the sum is the sum of the variances,

$$\mathrm{Var}\left(\mathbf{I}_{\mathrm{EE},i}^{[l]}\right) = \sum_{j=1}^{d} \mathrm{Var}\left(\mathrm{W}_{\mathrm{EE},ij}^{[l]} s_{\mathrm{E},j}^{[l-1]}\right). \qquad (33)$$

By further assuming that the weights $W_{\text{EE},ij}^{[l]}$ and input signals $s_{\text{E},j}^{[l-1]}$ are independently distributed, we can simplify this to

$$\text{Var}\left(\mathbf{I}_{\text{EE},i}^{[l]}\right) = d \cdot \text{Var}\left(\mathbf{W}_{\text{EE}}^{[l]} s_{\text{E}}^{[l-1]}\right) \tag{34}$$

$$= d\left(\mathbb{E}\left[(s_{\text{E}}^{[l-1]})^2\right]\text{Var}\left(\mathbf{W}_{\text{EE}}^{[l]}\right) + \text{Var}\left(s_{\text{E}}^{[l-1]}\right)\mathbb{E}^2\left[\mathbf{W}_{\text{EE}}^{[l]}\right]\right). \tag{35}$$

As established in the main text, we model the input spikes as an i.i.d. Bernoulli distribution with parameter $p$. Thus, $\mathbb{E}\left[s_{\text{E}}^{[l-1]}\right] = p$, $\mathbb{E}\left[(s_{\text{E}}^{[l-1]})^2\right] = p$, and $\text{Var}\left(s_{\text{E}}^{[l-1]}\right) = p(1-p)$. The weights $\mathbf{W}_{\text{EE}}^{[l]}$ are drawn from an exponential distribution with rate $\lambda^{[l]}$, for which $\mathbb{E}\left[\mathbf{W}_{\text{EE}}^{[l]}\right] = 1/\lambda^{[l]}$ and $\text{Var}\left(\mathbf{W}_{\text{EE}}^{[l]}\right) = 1/(\lambda^{[l]})^2$. Substituting these into Equation 35 yields:

$$\text{Var}\left(\mathbf{I}_{\text{EE},i}^{[l]}\right) = d\left(p \cdot \frac{1}{(\lambda^{[l]})^2} + p(1-p) \cdot \frac{1}{(\lambda^{[l]})^2}\right) \tag{36}$$

$$= \frac{d}{(\lambda^{[l]})^2}\left(p + p(1-p)\right) \tag{37}$$

$$= \frac{dp(2-p)}{(\lambda^{[l]})^2}. \tag{38}$$

Therefore, the standard deviation of $\mathbf{I}_{\text{EE},i}^{[l]}$ is:

$$\text{std}\left(\mathbf{I}_{\text{EE},i}^{[l]}\right) = \sqrt{\text{Var}\left(\mathbf{I}_{\text{EE},i}^{[l]}\right)} = \frac{\sqrt{dp(2-p)}}{\lambda^{[l]}}. \tag{39}$$

### D.4 SELECTION OF RATE PARAMETER $\lambda^{[l]}$

To ensure stable signal propagation at initialization, we set the standard deviation of the input currents to match that of the input spikes, i.e., $\text{std}\left(\mathbf{I}_{\text{EE},i}^{[l]}\right) = \text{std}\left(\mathbf{s}_{\text{E}}^{[l-1]}\right) = \sqrt{p(1-p)}$. Using the result from Equation 39 we have

$$\frac{\sqrt{dp(2-p)}}{\lambda^{[l]}} = \sqrt{p(1-p)}, \tag{40}$$

$$\lambda^{[l]} = \frac{\sqrt{dp(2-p)}}{\sqrt{p(1-p)}} = \sqrt{\frac{d(2-p)}{1-p}}. \tag{41}$$

### D.5 BACKPROPAGATION OF THE E-I CIRCUIT

In this section, we provide a detailed backpropagation derivation for the E-I circuit to theoretically justify the necessity of the gradient scaling on $\mathbf{W}_{\text{EI}}$.

We first re-write the forward pass formulation at time step $t$ for layer $l$:

$$\mathbf{I}_{\text{int}}^{[l]}[t] = \mathbf{g}_{\text{E}}^{[l]} \odot \frac{\mathbf{I}_{\text{EE}}^{[l]}[t] - \mathbf{I}_{\text{EI,sub}}^{[l]}[t]}{\mathbf{I}_{\text{EI,div}}^{[l]}[t]} + \mathbf{b}_{\text{E}}^{[l]}, \tag{42}$$

where the currents are defined as:

$$\mathbf{I}_{\text{EI,sub}}^{[l]}[t] = \mathbf{W}_{\text{EI}}^{[l]}\mathbf{s}_{\text{I}}^{[l]}[t], \tag{43}$$

$$\mathbf{I}_{\text{EI,div}}^{[l]}[t] = \mathbf{W}_{\text{EI}}^{[l]}(\mathbf{g}_{\text{I}}^{[l]} \odot \mathbf{s}_{\text{I}}^{[l]}[t]), \tag{44}$$

$$\mathbf{s}_{\text{I}}^{[l]}[t] \approx \mathbf{W}_{\text{IE}}^{[l]}\mathbf{s}_{\text{E}}^{[l-1]}[t]. \tag{45}$$

Note that here we use the linear approximation for $\mathbf{s}_{\text{I}}$ derived in Appendix D.1.

Let $\boldsymbol{\delta}^{[l]}[t] = \frac{\partial\mathcal{L}}{\partial\mathbf{I}_{\text{int}}^{[l]}[t]}$ be the error signal backpropagated from subsequent layers at time $t$. The gradients are accumulated over $T$ time steps. Below we provide the derivative of loss w.r.t. each trainable parameter in layer $l$ of the E-I circuit.

1. $\boldsymbol{W}_{\mathrm{EE}}^{[l]}$.

$$\frac{\partial \mathcal{L}}{\partial \boldsymbol{W}_{\mathrm{EE}}^{[l]}} = \sum_{t=1}^{T} \left[ \left( \boldsymbol{\delta}^{[l]}[t] \odot \frac{\mathbf{g}_{\mathrm{E}}^{[l]}}{\mathbf{I}_{\mathrm{EI,div}}^{[l]}[t]} \right) \left( \mathbf{s}_{\mathrm{E}}^{[l-1]}[t] \right)^{\top} \right]. \tag{46}$$

2. $\boldsymbol{W}_{\mathrm{EI}}^{[l]}$.

$\boldsymbol{W}_{\mathrm{EI}}^{[l]}$ contributes to both the subtractive and divisive pathways. Therefore, its gradient can be decomposed into subtractive and divisive components that correspond to the two types of inhibitory currents.

$$\frac{\partial \mathcal{L}}{\partial \boldsymbol{W}_{\mathrm{EI}}^{[l]}} = \sum_{t=1}^{T} \left[ \underbrace{\left( \boldsymbol{\delta}^{[l]}[t] \odot \frac{-\mathbf{g}_{\mathrm{E}}^{[l]}}{\mathbf{I}_{\mathrm{EI,div}}^{[l]}[t]} \right) \left( \mathbf{s}_{\mathrm{I}}^{[l]}[t] \right)^{\top}}_{\text{Subtractive Component}} \right.$$

$$\left. + \underbrace{\left( \boldsymbol{\delta}^{[l]}[t] \odot \frac{-\mathbf{g}_{\mathrm{E}}^{[l]} \odot \mathbf{I}_{\mathrm{balanced}}^{[l]}[t]}{\mathbf{I}_{\mathrm{EI,div}}^{[l]}[t] \odot \mathbf{I}_{\mathrm{EI,div}}^{[l]}[t]} \right) \left( \mathbf{g}_{\mathrm{I}}^{[l]} \odot \mathbf{s}_{\mathrm{I}}^{[l]}[t] \right)^{\top}}_{\text{Divisive Component}} \right], \tag{47}$$

where $\mathbf{I}_{\mathrm{balanced}}^{[l]}[t] = \mathbf{I}_{\mathrm{EE}}^{[l]}[t] - \mathbf{I}_{\mathrm{EI,sub}}^{[l]}[t]$.

3. $\boldsymbol{W}_{\mathrm{IE}}^{[l]}$.

The error propagates back through the inhibitory neurons.

$$\frac{\partial \mathcal{L}}{\partial \boldsymbol{W}_{\mathrm{IE}}^{[l]}} = \sum_{t=1}^{T} \left( \boldsymbol{\delta}_{\mathrm{I}}^{[l]}[t] \right) \left( \mathbf{s}_{\mathrm{E}}^{[l-1]}[t] \right)^{\top}, \tag{48}$$

where the error term $\boldsymbol{\delta}_{\mathrm{I}}$ combines gradients from both $\boldsymbol{W}_{\mathrm{EI}}^{[l]}$ pathways. Therefore,

$$\boldsymbol{\delta}_{\mathrm{I}}^{[l]}[t] = \boldsymbol{W}_{\mathrm{EI}}^{[l]\top} \left( \boldsymbol{\delta}^{[l]}[t] \odot \frac{-\mathbf{g}_{\mathrm{E}}^{[l]}}{\mathbf{I}_{\mathrm{EI,div}}^{[l]}[t]} \right) + \mathbf{g}_{\mathrm{I}}^{[l]} \odot \left[ \boldsymbol{W}_{\mathrm{EI}}^{[l]\top} \left( \boldsymbol{\delta}^{[l]}[t] \odot \frac{-\mathbf{g}_{\mathrm{E}}^{[l]} \odot \mathbf{I}_{\mathrm{balanced}}^{[l]}[t]}{\mathbf{I}_{\mathrm{EI,div}}^{[l]}[t] \odot \mathbf{I}_{\mathrm{EI,div}}^{[l]}[t]} \right) \right]. \tag{49}$$

4. $\mathbf{g}_{\mathrm{I}}^{[l]}$ This parameter modulates the inhibitory signals before they are projected by $\boldsymbol{W}_{\mathrm{EI}}^{[l]}$ for the divisive pathway. Therefore, the error signal propagates back through the divisive branch of $\boldsymbol{W}_{\mathrm{EI}}^{[l]}$.

$$\frac{\partial \mathcal{L}}{\partial \mathbf{g}_{\mathrm{I}}^{[l]}} = \sum_{t=1}^{T} \left[ \boldsymbol{W}_{\mathrm{EI}}^{[l]\top} \left( \boldsymbol{\delta}^{[l]}[t] \odot \frac{-\mathbf{g}_{\mathrm{E}}^{[l]} \odot \mathbf{I}_{\mathrm{balanced}}^{[l]}[t]}{\mathbf{I}_{\mathrm{EI,div}}^{[l]}[t] \odot \mathbf{I}_{\mathrm{EI,div}}^{[l]}[t]} \right) \right] \odot \mathbf{s}_{\mathrm{I}}^{[l]}[t]. \tag{50}$$

5. $\mathbf{g}_{\mathrm{E}}^{[l]}$.

$$\frac{\partial \mathcal{L}}{\partial \mathbf{g}_{\mathrm{E}}^{[l]}} = \sum_{t=1}^{T} \left( \boldsymbol{\delta}^{[l]}[t] \odot \frac{\mathbf{I}_{\mathrm{balanced}}^{[l]}[t]}{\mathbf{I}_{\mathrm{EI,div}}^{[l]}[t]} \right). \tag{51}$$

6. $\mathbf{b}_{\mathrm{E}}^{[l]}$.

$$\frac{\partial \mathcal{L}}{\partial \mathbf{b}_{\mathrm{E}}^{[l]}} = \sum_{t=1}^{T} \boldsymbol{\delta}^{[l]}[t]. \tag{52}$$

# E IMPLEMENTATION DETAILS

## E.1 CNNS WITH THE E-I CIRCUIT

In experiments, we mainly use CNNs like VGG and ResNet, where all convolutional layers and the fully connected classifier are implemented with the proposed E-I circuit (for output layer, we do not apply divisive inhibition to better stabilize the output logits). Here we clarify the implementation of the convolution layer used in our experiments. Specifically, $\boldsymbol{W}_{\mathrm{EE}}$ and $\boldsymbol{W}_{\mathrm{IE}}$ are standard $K \times K$ convolution kernels, while the lateral connection $\boldsymbol{W}_{\mathrm{EI}}$ is a $1 \times 1$ point-wise convolution kernel. In this context, the input dimension $d$ is defined as $d = C_{\mathrm{in}} \times K \times K$, where $C_{\mathrm{in}}$ is the number of channels of excitatory input, and the population sizes $n_{\mathrm{E}}$ and $n_{\mathrm{I}}$ are the number of excitatory and inhibitory output channels, respectively. This configuration allows inhibitory neurons to regulate excitatory neurons densely across the channel dimension while preserving the spatial structure.

## E.2 DYNAMIC INITIALIZATION

While our theoretical analysis under Bernoulli assumption provides basic ideas of E-I Init, we find it more effective and stable during training if we estimate statistics from training data at initialization, rather than formulating them with Bernoulli distribution parameter $p$ and manually set $p$. Similar to the theoretical analysis under the Bernoulli assumption above, we have

$$\lambda^{[l]} = \sqrt{\frac{d\left(\mathbb{E}\left[\left(\mathbf{s}_{\mathrm{E}}^{[l-1]}\right)^2\right] + \mathrm{Var}\left(\mathbf{s}_{\mathrm{E}}^{[l-1]}\right)\right)}{\mathrm{Var}\left(\mathbf{s}_{\mathrm{E}}^{[l-1]}\right)}}, \tag{53}$$

$$\mathbf{g}_{\mathrm{I}}^{[l]} = \sqrt{\frac{\left(\mathbb{E}\left[\left(\mathbf{s}_{\mathrm{E}}^{[l-1]}\right)^2\right] + \mathrm{Var}\left(\mathbf{s}_{\mathrm{E}}^{[l-1]}\right)\right)}{d\mathbb{E}^2\left[\mathbf{s}_{\mathrm{E}}^{[l-1]}\right]}}. \tag{54}$$

Therefore, by computing mean, second raw moment, and sample-wise variance only once at initialization, the model performs self-regulated E-I balance and gain control, stabilizing training without explicit normalization. Algorithm 1 summarizes the implementation of the proposed E-I init.

---

**Algorithm 1** E-I Init

---

**Require:** Input $\mathbf{X}$ of shape $(B, \dots)$ (first batch), input dimension $d$, inhibitory neuron count $n_{\mathrm{I}}$.
**Ensure:** Initialized parameters $\boldsymbol{W}_{\mathrm{EE}}, \boldsymbol{W}_{\mathrm{IE}}, \boldsymbol{W}_{\mathrm{EI}}, \mathbf{g}_{\mathrm{I}}, \mathbf{g}_{\mathrm{E}}, \mathbf{b}_{\mathrm{E}}$.
1: **procedure** EI-INIT($\mathbf{X}, d, n_{\mathrm{I}}$)
                                         ▷ **Step 1: Estimate input statistics from the first batch $\mathbf{X}$**
2:       mean $\leftarrow \mathbf{X}$.mean()
3:       var $\leftarrow \mathbf{X}$.var(dim = 0).mean()
4:       moment $\leftarrow (\mathbf{X}^2)$.mean()
                                           ▷ **Step 2: Calculate the rate parameter based on statistics**
5:       exp_scale $\leftarrow \sqrt{\frac{\mathrm{var}}{d \cdot (\mathrm{moment}+\mathrm{var})}}$                       ▷ exp_scale $= 1/\lambda$
6:       gain_I $\leftarrow \frac{1}{\sqrt{d}} \cdot \frac{\sqrt{\mathrm{moment}+\mathrm{var}}}{\mathrm{mean}}$                 ▷ Initial value for $\mathbf{g}_{\mathrm{I}}$
                                            ▷ **Step 3: Initialize trainable parameters**
7:       $\mathrm{W}_{\mathrm{EE}} \sim$ EXPONENTIAL(scale = exp_scale)
8:       $\mathrm{W}_{\mathrm{IE}} \sim$ EXPONENTIAL(scale = exp_scale)
9:       $\mathrm{W}_{\mathrm{EI}} \leftarrow \frac{1}{n_{\mathrm{I}}}$
10:      $\mathbf{g}_{\mathrm{I}} \leftarrow$ gain_I
11:      $\mathbf{g}_{\mathrm{E}} \leftarrow \mathbf{1}$
12:      $\mathbf{b}_{\mathrm{E}} \leftarrow \mathbf{0}$
13: **end procedure**

---

### E.3 ADAPTIVE STABILIZATION WITH STE

Algorithm 2 demonstrates the full procedure of our proposed adaptive stabilization of divisive inhibition mechanism, including the backward pass with STE.

---

**Algorithm 2** E-I Prop

---

**Require:** Input $\mathbf{X}$ of shape $(B, \dots)$, where $B$ is the batch size.
**Ensure:** Output $\mathbf{X}_{\text{out}}$ with zeros adaptively replaced.
 1: **procedure** ADAPTIVESTABILIZATION($\mathbf{X}$)
 2:     **if** $\mathbf{X}$ contains no zero values **then**
 3:         **return** $\mathbf{X}$
 4:     **end if**
                            ▷ **Step 1: Replace zeros with second minimum**
 5:     $\mathbf{M} \leftarrow (\mathbf{X} == 0)$                    ▷ Create a boolean mask for all zero locations
 6:     $\mathbf{X}_{\text{tmp}} \leftarrow \mathbf{X}$
 7:     $\mathbf{X}_{\text{tmp}}[\mathbf{M}] \leftarrow \infty$                       ▷ Temporarily replace zeros with infinity
 8:     **for** each sample $i$ from 1 to $B$ **do**
 9:         $s_i \leftarrow \min(\mathbf{X}_{\text{tmp}}[i])$          ▷ Find the smallest positive value of the original sample
10:         $\mathbf{S}[i] \leftarrow s_i$
11:     **end for**
12:     $\mathbf{X}_{\text{fwd}} \leftarrow \text{where}(\mathbf{M}, \mathbf{S}, \mathbf{X})$       ▷ Replace zeros with the smallest positive value of the sample
                                  ▷ **Step 2: Construct the final output with STE**
13:     $\mathbf{X}_{\text{out}} \leftarrow \text{detach}(\mathbf{X}_{\text{fwd}}) + (\mathbf{X} - \text{detach}(\mathbf{X}))$          ▷ STE via the detach trick
14:     **return** $\mathbf{X}_{\text{out}}$
15: **end procedure**

---

## F SUPPLEMENTARY RESULTS AND EXPERIMENT DETAILS

### F.1 EMPIRICAL ANALYSIS OF GRADIENT FLOW AND SCALING ROBUSTNESS

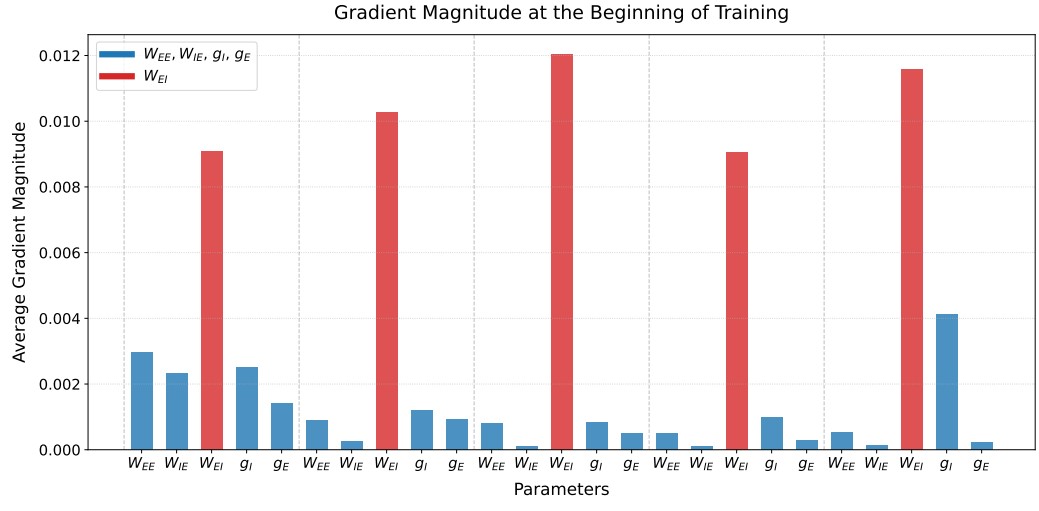

Figure 5: Empirical analysis of gradient norms at initialization for convolutional layers in VGG-8, without gradient scaling. The gradients for $\mathbf{W}_{\text{EI}}$ (**red**) are orders of magnitude larger than those for $\mathbf{W}_{\text{EE}}$, $\mathbf{W}_{\text{IE}}$, and gain parameters (**blue**), consistent with the theoretical analysis.

Figure 5 visualizes the magnitudes of gradient norms for all trainable parameters across the convolutional layers of VGG-8 at the first training iteration (without gradient scaling). Consistent with our

theoretical derivation in Appendix D.5, the gradient norms for $W_{\text{EI}}$ are disproportionately larger than those of other parameters. This imbalance stems primarily from the divisive operation, which introduces a term proportional to $1/\left(\mathbf{I}_{\text{EI,div}}^{[l]}[t] \odot \mathbf{I}_{\text{EI,div}}^{[l]}[t]\right)$ in the gradient, leading to quadratic amplification when the denominator is small. While the gradients for $W_{\text{IE}}^{[l]}$ and $\mathbf{g}_{\text{I}}^{[l]}$ also contain this term, they are implicitly dampened since the gradients backpropagate through $W_{\text{EI}}^{[l]}$, which acts as an averaging filter due to its deterministic initialization of $1/n_{\text{I}}^{[l]}$. In contrast, the gradient for $W_{\text{EI}}^{[l]}$ lacks such averaging mechanism and is instead directly proportional to the inhibitory activity $\mathbf{s}_{\text{I}}^{[l]}$. Consequently, the gradient magnitude of $W_{\text{EI}}^{[l]}$ is driven by the input spikes $\mathbf{s}_{\text{I}}^{[l]}[t] \approx W_{\text{IE}}^{[l]}\mathbf{s}_{\text{E}}^{[l-1]}[t]$, scaling linearly with the input dimension $d$.

To counteract this amplification, we choose the scaling factor as $1/d$. To verify the robustness of scaling factor choice, a sensitivity analysis is conducted on VGG-8 (CIFAR-10) by varying the scaling factor from $1/\sqrt{d}$ to $1/d^2$ (see Table 3). The result demonstrates that our method is stable across a broad range of scaling factors (e.g., $1/\sqrt{d}$ and $1/d$ yield comparable performance), whereas removing the scaling immediately leads to collapse.

Table 3: Sensitivity analysis of gradient scaling factor with VGG-8 on CIFAR-10.

| Scaling Factor | Top-1 Accuracy (%) |
|---|---|
| No Scaling | Collapsed |
| $1/\sqrt{d}$ | 86.87 |
| $1/d$ (Default) | **86.88** |
| $1/d^2$ | 86.33 |

## F.2 COMPARISON WITH E-I ANNS

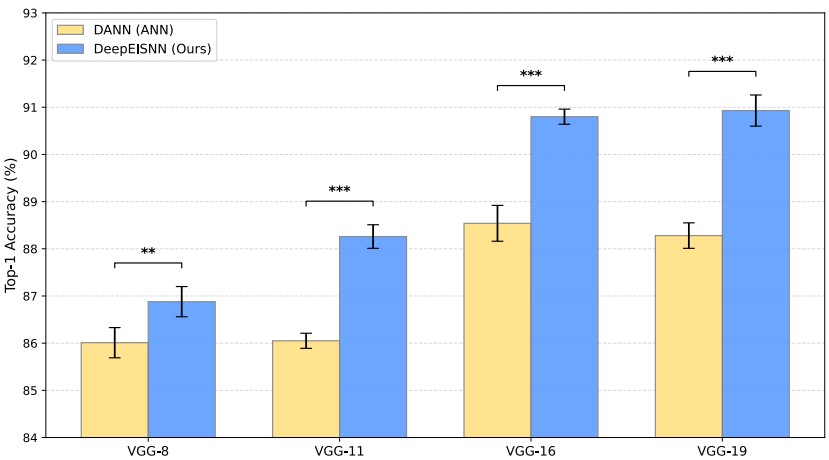

Figure 6: Comparison between our method and DANN on CIFAR-10. Error bars denote the standard deviation over multiple independent runs. Statistical significance between the two methods is indicated by asterisks (** $p < 0.01$, *** $p < 0.001$).

Comparison with DANN (Cornford et al., 2021) highlights the advantage of our method. Figure 6 shows that our method consistently and significantly outperforms DANN across all tested VGG architectures. Statistical analysis confirms that these improvements are statistically significant ($p < 0.01$ for VGG-8 and $p < 0.001$ for deeper models). Notably, the advantage of our method increases as the network depth increases. The performance gap widens from $0.87\%$ on VGG-8 to $2.65\%$ on VGG-19. This trend strongly suggests that our proposed mechanisms, E-I Init and E-I Prop, are

more effective at preserving stable signal propagation and facilitating effective learning in very deep architectures.

### F.3 DISTRIBUTIONS OF BN OUTPUTS

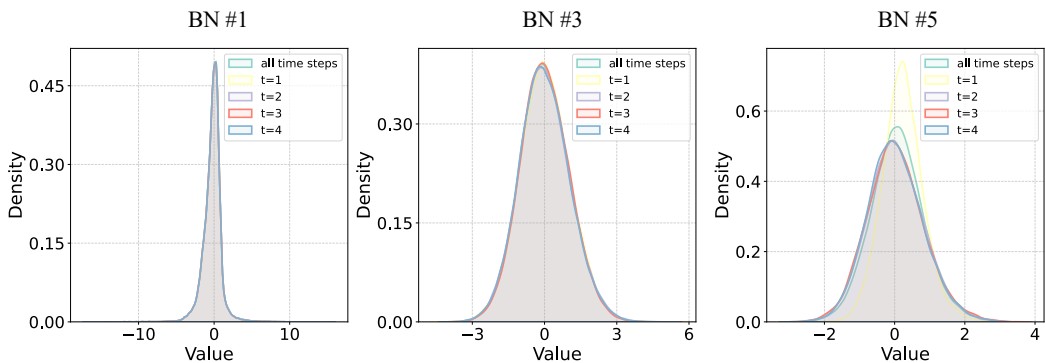

Figure 7: Distributions of the outputs in the first, third and fifth BN layers after training.

Figure 7 demonstrates output distributions of BN layers after training, which are all Gaussian-like and zero-centered.

### F.4 VISUALIZATION OF E-I INTERACTION

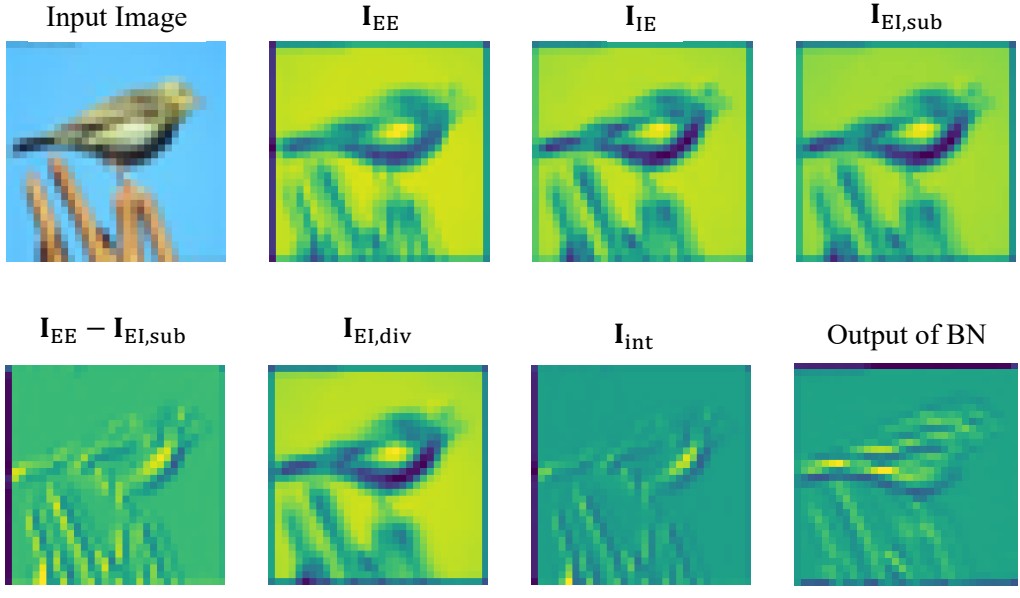

Figure 8: Comparison between feature maps of the first E-I circuit layer in our model and the feature map of the first BN layer in SNN with vanilla BN after training.

A visual comparison of the feature maps suggests that our E-I circuit and vanilla BN forces network to focus on different feature representations. As shown in Figure 8, our E-I circuit produces a feature map where activations are concentrated along the object's contours, indicating a learned focus on feature edges and boundaries. In contrast, a standard BN layer may preserve a dense spatial output of

its preceding convolution, normalizing the representation of the feature's overall shape and texture rather than isolating its boundaries.

### F.5 COMPUTATIONAL OVERHEAD ANALYSIS

To quantify the computational cost associated with ensuring biological fidelity, we measure the training time (per epoch) and peak GPU memory usage on a single NVIDIA GeForce RTX 4090. We compare our DeepEISNN with SNNs with vanilla BN across various architectures. The results are summarized in Table 4.

Table 4: Computational overhead ($T = 4$, batch size=128, single GPU).

| Arch. | Metric | SNN (Baseline) | DeepEISNN (Ours) | Factor |
|---|---|---|---|---|
| VGG-8 | Time/Epoch | 7.3s | 18.1s | 2.48× |
|  | Memory | 2394 MB | 2684 MB | 1.12× |
| VGG-11 | Time/Epoch | 9.5s | 23.1s | 2.43× |
|  | Memory | 2392 MB | 2886 MB | 1.21× |
| VGG-16 | Time/Epoch | 16.9s | 42.2s | 2.50× |
|  | Memory | 3544 MB | 5072 MB | 1.43× |
| VGG-19 | Time/Epoch | 19.4s | 47.6s | 2.45× |
|  | Memory | 3734 MB | 5402 MB | 1.45× |
| ResNet-18 | Time/Epoch | 38.8s | 90.0s | 2.32× |
|  | Memory | 5918 MB | 9302 MB | 1.57× |

As shown in Table 4, our method introduces a computational overhead of approximately $2.3\times \sim 2.5\times$ in training time and $1.1\times \sim 1.6\times$ in GPU memory usage. This increase is an expected and necessary trade-off for the E-I circuit with a 4:1 excitatory-to-inhibitory ratio. Unlike standard SNNs that utilize a single synaptic weight matrix per layer, our framework explicitly models three distinct synaptic projections ($W_{EE}, W_{IE}, W_{EI}$) and maintains an additional inhibitory population. Importantly, this overhead scales linearly with network size, ensuring tractability for deep architectures. Despite the increased per-epoch cost, our method demonstrates robust convergence comparable to BN-equipped baselines (as evidenced by the competitive accuracy in Table 1), thereby enabling normalization-free learning using biologically grounded mechanisms.

### F.6 EXPERIMENT DETAILS

Code is implemented using the PyTorch framework and run on NVIDIA GeForce RTX 4090 GPUs.

**Network architectures.** We employ standard backbones, including VGG-8/11/16/19 (Simonyan & Zisserman, 2015) and ResNet-18 (He et al., 2016; Fang et al., 2021a). In these architectures, the standard convolutional blocks (Conv-BN-LIF) and linear classifiers are replaced by our proposed E-I circuit. To construct a lightweight classifier, we apply global average pooling (GAP) before the linear readout layer. VGG-8 is utilized primarily for ablation studies on CIFAR-10, while deeper models (VGG-16/19, ResNet-18) are employed for SOTA comparisons and large-scale benchmarks.

**Data preprocessing.** Our method is validated on multiple datasets, including CIFAR-10/100 (Krizhevsky, 2009), CIFAR10-DVS (Li et al., 2017), DVS-Gesture (Amir et al., 2017), and TinyImageNet (Le & Yang, 2015), using their standard train and validation splits. We apply distinct data augmentation strategies on different datasets. Specifically, for CIFAR-10 and CIFAR-100, we employ random cropping with a size of $32 \times 32$ (padding of 4 pixels), random horizontal flipping, and cutout. For TinyImageNet, images are downsampled to $32 \times 32$, and augmentations include random resized cropping, random horizontal flipping, color jittering, and cutout. Regarding neuromorphic datasets, both CIFAR10-DVS and DVS-Gesture are resized to a spatial resolution of $48 \times 48$. The training pipeline for these event-based datasets includes random resized cropping and random horizontal flipping. Additionally, we apply random temporal deletion specifically for the DVS-Gesture dataset to enhance temporal robustness.

**Global configuration.** All models are trained for 300 epochs. The optimization is performed using SGD with a momentum of 0.9 and a weight decay of 0.0005, together with a cosine annealing learning rate scheduler combined with an initial linear warm-up. The ratio of excitatory to inhibitory neurons is fixed at 4:1 across all layers. No dropout is applied. We use the standard cross-entropy loss. The final prediction is obtained by averaging the output logits of the classifier across all simulation time steps before computing the loss. For performance evaluation, we report the best top-1 accuracy achieved on the validation set throughout the training process.

**Task-specific configuration.** To accommodate the varying complexities and temporal dynamics of different datasets and network architectures, we finetune some hyperparameters for each specific task. The detailed task-specific configurations are summarized in Table 5.

Table 5: Task-specific hyperparameter configurations.

| Dataset | Architecture | Batch Size | Time Steps | Peak LR | Warm-up Epochs |
|---------|-------------|------------|------------|---------|----------------|
| CIFAR-10 | VGG-8/11 | 128 | 4 | 0.002 | 10 |
|          | VGG-16/19, ResNet-18 | 128 | 4 | 0.001 | 30 |
| CIFAR-100 | VGG-16/19 | 128 | 4 | 0.001 | 30 |
| TinyImageNet | ResNet-18 | 128 | 4 | 0.003 | 10 |
| CIFAR10-DVS | VGG-8/11 | 32 | 10 | 0.001 | 10 |
| DVS-Gesture | VGG-8 | 32 | 16 | 0.001 | 10 |

