# OpenReview forum: "Training Deep Normalization-Free Spiking Neural Networks with Lateral Inhibition"
_ICLR.cc/2026/Conference — ICLR 2026 Poster_

### Official Review · Reviewer_s6FP · 2025-10-23

**Soundness:** 3
**Presentation:** 2
**Contribution:** 2
**Rating:** 4
**Confidence:** 4

**Summary:**

The paper proposes a biologically-inspired framework to enable the stable training of deep Spiking Neural Networks (SNNs) without explicit normalization layers. This is achieved by introducing a lateral inhibition circuit composed of distinct excitatory and inhibitory populations as a substitute for standard normalization. To this end, the authors have developed a comprehensive suite of techniques, including a principled initialization scheme (E-I Init) and a stabilization method for propagation (E-I Prop).

**Strengths:**

This work addresses a critical challenge in the field of neuromorphic computing that training deep SNNs while preserving biological plausibility with a novel and well-conceived solution. The proposed E-I dynamic regulation mechanism is compelling, and the thorough ablation studies convincingly demonstrate the framework's effectiveness and the necessity of its individual components.

**Weaknesses:**

1.The simplification of the inhibitory neuron model to a static, ReLU-like unit undermines the network's temporal dynamics. While computationally convenient, its validity is questionable, and it may introduce significant errors, especially when the number of timesteps (T) is small.

2.The 1/d gradient scaling for the inhibitory weights W_EI appears to be an ad-hoc choice. The paper lacks a theoretical analysis or extensive empirical validation explaining why this specific scaling factor is optimal, which may raise concerns about its generalizability.

3.The paper is missing a direct performance comparison with state-of-the-art (SOTA) SNNs that use Batch Normalization (BN) within the same architectures. This weakens the claim of achieving "competitive performance."

4.All experiments are conducted on static image classification tasks. To truly validate the framework's temporal processing capabilities, it should be evaluated on event-based datasets (e.g., DVS-Gesture) or other sequential tasks where SNNs are expected to excel.

5.As depicted in Figure 2, the model incorporates both subtractive and divisive inhibition. However, the manuscript (e.g., around Equation 14) fails to provide a clear mathematical formula describing how these two operations are combined. This ambiguity severely hinders the method's reproducibility.

**Questions:**

1.Does the proposed method alter the standard forward-pass rules of SNNs? Can the operations within the E-I circuit, particularly the divisive inhibition, be efficiently deployed on existing neuromorphic hardware platforms?

2.A core motivation of this work is the biological implausibility of BN. However, during inference, BN parameters can be folded into the weights of the preceding convolutional or fully-connected layers, meaning no standalone BN operation exists in the final network. Could you please elaborate on why BN is still considered a critical barrier to the biological plausibility of SNNs in this context?

---

> ### Author Response · Authors · 2025-11-24
> **Response to Reviewer s6FP [1/7]**
>
> Dear Reviewer s6FP,
>
> We sincerely thank you for your clear and insightful review, which covers many aspects that we are willing to further investigate and discuss about our work. Below is our response to each of your concerns.
>
> > ***Weakness 1:*** *The simplification of the inhibitory neuron model to a static, ReLU-like unit undermines the network's temporal dynamics. While computationally convenient, its validity is questionable, and it may introduce significant errors, especially when the number of timesteps (T) is small.*
>
> We appreciate this insightful comment on the temporal dynamics of our model. Here we'd like to further explain that the approximation will not lead to significant error, and it is indeed a mathematically rigorous approximation and a necessary design choice for discrete-time simulation involving multiple time scales. Below we justify our design for the inhibitory neurons from four perspectives.
>
> **1. Mathematical Justification: The approximation error is independent of the total time steps $T$.**
>
> First we prove formally that the LIF model for inhibitory neurons effectively converges to a stateless unit whose current state is only determined by the current input, as long as $\tau_\mathrm I\ll\Delta t$. Here $\tau_\mathrm I$ is the membrane constant of the inhibitory neurons and $\Delta t$ is the simulation time step size. Importantly, the derivation shows that the validity of the approximation depends on the ratio $\Delta t / \tau_{\mathrm I}$, rather than the total simulation duration $T$.
>
> The sub-threshold dynamics of a LIF inhibitory neuron is given as follows:
>
> $$\tau_{\mathrm I}\frac{\mathrm d \mathbf u_{\mathrm I}^{[l]}(t)}{\mathrm dt}=-\left(u_{\mathrm I}^{[l]}(t)-u_{\mathrm{I, rest}}\right)+\mathbf I_{\mathrm {I}}^{[l]}(t),$$
>
> with all notations consistent with the manuscript. Assuming $u_{\mathrm{I, rest}}=0$ and constant input $\mathbf I_0$ over the duration $\Delta t$ starting from $t$, the analytical solution for the membrane potential at time $t+\Delta t$ is
>
> $$\mathbf u_{\mathrm I}^{[l]}(t+\Delta t)=\mathbf u_{\mathrm I}^{[l]}(t)e^{-\frac{\Delta t}{\tau _\mathrm{I}}} + \mathbf I_0\left(1-e^{-\frac{\Delta t}{\tau _{\mathrm I}}}\right).$$
>
> In our discrete-time simulation, excitatory neurons have $\tau_{\mathrm E}=2$, which has the same order of magnitude of $\Delta t = 1$. In contrast, inhibitory neurons are modeled as fast-spiking (FS) neurons, which biologically have a much smaller time constant than that of excitatory neurons [1, 2], i.e., $\tau_{\mathrm I}\ll \tau_{\mathrm E} = \Theta(\Delta t)$, where $\Theta$ denotes the same order of magnitude. Therefore, the ratio $\frac{\Delta t}{\tau _{\mathrm I}}$ becomes very large and thus $e^{-\frac{\Delta t}{\tau _\mathrm{I}}}\rightarrow0$ under this condition. By substituting the above derivation into the potential update rule, we have
>
> $$\mathbf u_{\mathrm I}^{[l]}(t+1)=\mathbf I_0,$$
>
> which means that the inhibitory neurons can be effectively modeled as stateless units as long as $\tau_{\mathrm I}\ll \Delta t$, since the membrane potential reaches steady state (totally determined by the input current) almost instantaneously within a single time step. This justifies our approximation in discrete-time simulation.
>
> Furthermore, since our simulation time step is much longer than the dynamics of inhibitory neurons, the potential tracks the input instantly and the neurons can fire multiple times at a single time step. By applying a threshold of 1 and a soft-reset mechanism to inhibitory neurons, we arrive at the final ReLU-like approximation in the manuscript:
>
> $$\mathbf s_{\mathrm I}^{[l]}(t+1)=\left\lfloor\max\left(0,\mathbf I_0\right)\right\rfloor = \left\lfloor\mathrm{ReLU}(\mathbf I_0)\right\rfloor\approx\mathrm{ReLU}(\mathbf I_0).$$
>
> From above analysis we can confirm that no accumulation of approximation error w.r.t. total time steps $T$. And the validity of the model is strictly determined by the fact $\tau_{\mathrm I}\ll \Delta t$, making it reasonable even for short simulations (when $T$ is small).

---

> ### Author Response · Authors · 2025-11-24
> **Response to Reviewer s6FP [2/7]**
>
> **2. Empirical Verification: Performance remains robust across varying $T$.**
>
> To empirically verify the mathematical derivation above and address your concern regarding potential errors when the number of timesteps ($T$) is small, we conduct experiments with VGG-8 on CIFAR-10, varying T from the extreme case of $T=1$ to larger $T=16$ while keeping all other hyperparameters and random seed fixed. The results are shown in Table R1.
>
> **Table R1: Performance of VGG-8 on CIFAR-10 using different simulation time steps.**
>
> |Time Steps (T)|Top-1 Acc. (%)|
> | :--- | :--- |
> |1|84.97
> |2|85.86
> |4 (Default)|87.43
> |8|87.55
> |16|86.91
>
> As shown in Table R1, the model maintains a reasonable accuracy of 84.97% even when $T=1$. Although there is a performance drop compared to the default $T=4$ (87.43%), this is theoretically expected due to the reduction in the temporal integration duration of excitatory neurons, rather than artifacts from the inhibitory approximation. Crucially, performance remains stable and competitive across the entire range up to $T=16$ (86.91%).
>
> This empirical result challenges the hypothesis that the simplified inhibitory neurons would introduce significant errors in short simulations. Instead, it indicates that the instantaneous gain control provided by our inhibitory model effectively stabilizes the network even without temporal integration. These results align well with our mathematical derivation.
>
> **3. Biological Fidelity: The approximation captures the rapid kinetics of FS interneurons.**
>
> As mentioned in the manuscript, the inhibitory neurons in our model represent FS interneurons (usually PV+ neurons) physiologically, which are characterized by rapid membrane kinetics, short time constants, and less refractoriness, compared to excitatory pyramidal neurons [1, 2]. In biological circuits, these interneurons operate as rapid signaling units to provide in-time feedforward/feedback inhibition. By modeling them as stateless units, our approach effectively captures their functional role as quick responsers to stimulation, rather than slow integrators.
>
> **4. Functional Necessity: Instantaneous inhibition is necessary for stable timestep-wise gain control in our model.**
>
> In our method, the inhibitory pathway is designed to provide a biological alternative to explicit normalization. Functionally, the lateral inhibition must provide gain control that is synchronized with the excitatory input. If we model inhibitory neurons like excitatory neurons (with larger membrane constant), the inhibitory current will become an integration of past activity rather than the current input intensity. This temporal latency between excitation and inhibition will cause a mismatch in the divisive operation, potentially leading to network oscillations rather than effective gain control. Therefore, the instantaneous approximation is functionally reasonable and works well in our framework, ensuring that inhibition tracks and regulates excitation without phase lag to maintain network's stability.
>
> [1] Hijazi, S. *et al.*, Fast-spiking parvalbumin-positive interneurons in brain physiology and Alzheimer’s disease. *Mol. Psychiatry*, 2023.
>
> [2] Milicevic, KD. *et al.*, Physiological features of parvalbumin-expressing GABAergic interneurons contributing to high-frequency oscillations in the cerebral cortex. *Current Research in Neurobiology*, 2024.

---

> ### Author Response · Authors · 2025-11-24
> **Response to Reviewer s6FP [3/7]**
>
> > ***Weakness 2:*** *The 1/d gradient scaling for the inhibitory weights W_EI appears to be an ad-hoc choice. The paper lacks a theoretical analysis or extensive empirical validation explaining why this specific scaling factor is optimal, which may raise concerns about its generalizability.*
>
> We sincerely thank you for your attention to the details of our methodology. Here, we clarify that the scaling factor is not an ad-hoc choice applicable only to specific models or datasets. We justify our design from mathematical analysis of the backward pass.
>
> The forward pass at time $t$ is formulated as
>
> $$\mathbf{I}\_{\mathrm{int}}^{[l]}[t] = \mathbf{g}\_{\mathrm{E}}^{[l]} \odot \frac{\mathbf{I}\_{\mathrm{EE}}^{[l]}[t] - \mathbf{I}\_{\mathrm{EI,sub}}^{[l]}[t]}{\mathbf{I}\_{\mathrm{EI,div}}^{[l]}[t]} + \mathbf{b}\_{\mathrm{E}}^{[l]}, $$
>
> where $\mathbf{I}\_{\mathrm{EI,sub}}^{[l]}[t] = \mathbf{W}\_{\mathrm{EI}}^{[l]}\mathbf{s}\_{\mathrm{I}}^{[l]}[t]$, $\mathbf{I}\_{\mathrm{EI,div}}^{[l]}[t] = \mathbf{W}\_{\mathrm{EI}}^{[l]}(\mathbf{g}\_{\mathrm{I}}^{[l]} \odot \mathbf{s}\_{\mathrm{I}}^{[l]}[t])$ and $\mathbf{s}\_{\mathrm{I}}^{[l]}[t]=\max\left(0, \mathbf{W}\_{\mathrm{IE}}^{[l]} \mathbf{s}\_{\mathrm{E}}^{[l-1]}[t]\right)=\mathbf{W}\_{\mathrm{IE}}^{[l]} \mathbf{s}\_{\mathrm{E}}^{[l-1]}[t]$.
>
> To better support our analysis, we have provided the full backpropagation derivation in the Appendix of the revised manuscript. Below, we list the gradients of the loss with respect to the $\mathbf W_{\mathrm {EE}}^{[l]}$, $\mathbf W_{\mathrm {IE}}^{[l]}$ and $\mathbf W_{\mathrm {EI}}^{[l]}$ to better explain why we only need to scale gradients of $\mathbf W_{\mathrm {EI}}^{[l]}$ and why the scaling factor is chosen to be $1/d$.
>
> Let $\boldsymbol{\delta}^{[l]}[t] = \frac{\partial \mathcal{L}}{\partial \mathbf{I}\_{\mathrm{int}}^{[l]}[t]}$ be the error signal backpropagated from the subsequent layers at time $t$. Assuming the gradients of the weights are accumulated over all time steps $T$ (BPTT), then
>
> $$
> \frac{\partial \mathcal{L}}{\partial \mathbf{W}\_{\mathrm{EE}}^{[l]}} = \sum\_{t=1}^{T} \left[\left(\boldsymbol{\delta}^{[l]}[t] \odot \frac{\mathbf{g}\_{\mathrm{E}}^{[l]}}{\mathbf{I}\_{\mathrm{EI,div}}^{[l]}[t]} \right) \left(\mathbf{s}\_{\mathrm{E}}^{[l-1]}[t]\right)^{\top}\right],
> $$
>
> $$
> \frac{\partial \mathcal{L}}{\partial \mathbf{W}\_{\mathrm{EI}}^{[l]}} = \sum\_{t=1}^{T} \left[ \underbrace{\left( \boldsymbol{\delta}^{[l]}[t] \odot \frac{-\mathbf{g}\_{\mathrm{E}}^{[l]}}{\mathbf{I}\_{\mathrm{EI,div}}^{[l]}[t]} \right) \left(\mathbf{W}\_{\mathrm{IE}}^{[l]} \mathbf{s}\_{\mathrm{E}}^{[l-1]}[t]\right)^{\top}}\_{\text{Subtractive Component}} + \underbrace{\left( \boldsymbol{\delta}^{[l]}[t] \odot \frac{-\mathbf{g}\_{\mathrm{E}}^{[l]} \odot \mathbf{I}\_{\mathrm{balanced}}^{[l]}[t]}{\left(\mathbf{I}\_{\mathrm{EI,div}}^{[l]}[t]\right)^2} \right) \left(\mathbf{g}\_{\mathrm{I}}^{[l]} \odot \mathbf{W}\_{\mathrm{IE}}^{[l]} \mathbf{s}\_{\mathrm{E}}^{[l-1]}[t]\right)^{\top}}\_{\text{Divisive Component}} \right],
> $$
>
> $$
> \frac{\partial \mathcal{L}}{\partial \mathbf{W}\_{\mathrm{IE}}^{[l]}} = \sum\_{t=1}^{T} \left( \underbrace{\mathbf{W}\_{\mathrm{EI}}^{[l]\top} \left( \boldsymbol{\delta}^{[l]}[t] \odot \frac{-\mathbf{g}\_{\mathrm{E}}^{[l]}}{\mathbf{I}\_{\mathrm{EI,div}}^{[l]}[t]} \right)\left(\mathbf{s}\_{\mathrm{E}}^{[l-1]}[t]\right)^{\top}}\_{\text{Subtractive Component}} + \underbrace{\mathbf{g}\_{\mathrm{I}}^{[l]} \odot \left[ \mathbf{W}\_{\mathrm{EI}}^{[l]\top} \left( \boldsymbol{\delta}^{[l]}[t] \odot \frac{-\mathbf{g}\_{\mathrm{E}}^{[l]} \odot \mathbf{I}\_{\mathrm{balanced}}^{[l]}[t]}{\left(\mathbf{I}\_{\mathrm{EI,div}}^{[l]}[t]\right)^2} \right) \right]\left(\mathbf{s}\_{\mathrm{E}}^{[l-1]}[t]\right)^{\top}}\_{\text{Divisive Component}} \right),
> $$
>
> where $\mathbf I\_{\mathrm{balanced}}^{[l]}[t]=\mathbf{I}\_{\mathrm{EE}}^{[l]}[t] - \mathbf{I}\_{\mathrm{EI,sub}}^{[l]}[t]$.

---

> ### Author Response · Authors · 2025-11-24
> **Response to Reviewer s6FP [4/7]**
>
> Using results above, we first explain why we specifically scale the gradient of $\mathbf{W}\_{\mathrm{EI}}$. Comparing the gradient structures, $\mathbf{W}\_{\mathrm{EE}}$ is scaled by the factor $1/\mathbf{I}\_{\mathrm{EI,div}}$, whereas the gradient for $\mathbf{W}\_{\mathrm{EI}}$ and $\mathbf{W}\_{\mathrm{IE}}$ contains the term $1/(\mathbf{I}\_{\mathrm{EI,div}})^2$ in its divisive component. This quadratic term implies that when the inhibitory current is small, the gradient for $\mathbf{W}\_{\mathrm{EI}}$ and $\mathbf{W}\_{\mathrm{IE}}$ is amplified by orders of magnitude compared to $\mathbf{W}\_{\mathrm{EE}}$. This numerical sensitivity is identified as the primary source of training instability.
>
> However, for $\mathbf{W}\_{\mathrm{IE}}$, the error signal backpropagates through $\mathbf{W}\_{\mathrm{EI}}$. Since the elements of $\mathbf{W}\_{\mathrm{EI}}$ are initialized deterministically to $1/n\_{\mathrm{I}}$ (according to our E-I Init) and their growth is strictly constrained by the gradient scaling during training, the backward pass of gradient through $\mathbf{W}\_{\mathrm{EI}}$ effectively acts as an averaging operation. This naturally dampens the magnitude of the error signal reaching $\mathbf{W}\_{\mathrm{IE}}$, preventing gradient explosion.
>
> In contrast, for $\mathbf{W}\_{\mathrm{EI}}$, there is no such averaging mechanism to attenuate the gradient. Instead, the gradient is directly proportional to the inhibitory activity $\mathbf{s}\_{\mathrm{I}}^{[l]}[t]$ (derived from $\mathbf{W}\_{\mathrm{IE}}^{[l]} \mathbf{s}\_{\mathrm{E}}^{[l-1]}[t]$). Since this term involves aggregating inputs of $d$ features, its expected magnitude naturally scales linearly with the dimension $d$. This amplification, combined with the inherent numerical instability of the divisive operation, results in large gradients, thereby necessitating the explicit $1/d$ scaling to maintain stability.
>
> Empirical analysis (Figure 5 of the manuscript) also reveals that the gradient of $\mathbf{W}\_{\mathrm{EI}}$ is disproportionately larger than that of the other two weights, which is consistent with our mathematical analysis. Besides, we check the performance of VGG-8 on CIFAR-10 with scaling factors of different magnitudes, as shown in Table R2.
>
> **Table R2: Performance with different scaling factors.**
> |Scaling Factor|Top-1 Acc. (%)|
> |:---|:---|
> |No scaling|Collapsed|
> |$1/\sqrt{d}$|86.87|
> |$1/d$ (default)|86.88|
> |$1/d^2$|86.33|
>
> In summary, our theoretical analysis and empirical evidence consistently indicate that the scaling factor could be of the order **$\Theta(1/d)$**. Crucially, the results in Table R2 demonstrate that our method is robust to the specific choice of the scaling factor.
>
> > ***Weakness 3:*** *The paper is missing a direct performance comparison with state-of-the-art (SOTA) SNNs that use Batch Normalization (BN) within the same architectures. This weakens the claim of achieving "competitive performance."*
>
> > ***Weakness 4:*** *All experiments are conducted on static image classification tasks. To truly validate the framework's temporal processing capabilities, it should be evaluated on event-based datasets (e.g., DVS-Gesture) or other sequential tasks where SNNs are expected to excel.*
>
> Thanks for your attention to the performance evaluation. We agree that expanding the comparison scope to include both SOTA BN-equipped methods and event-based tasks significantly strengthens the paper. We have updated Table 1 in the revised manuscript (please refer to the Response to All Reviewers for the full table) and revised the Related Work section to include more BN-related literature.
>
> While we acknowledge a slight performance gap (approximately 2–3%) compared to BN-equipped baselines on static images, our method consistently outperforms other normalization-free and biologically constrained approaches. This confirms that our framework offers one of the best solutions within the domain of biologically plausible learning.
>
> Furthermore, our method performs well on event-based datasets, reaching 94.86% on DVS-Gesture and 77.66% on CIFAR10-DVS. Notably, our method surpasses several BN-equipped baselines on CIFAR10-DVS, demonstrating distinct advantages in processing asynchronous event streams.
>
> In summary, our results demonstrate that our method is superior among existing normalization-free or biologically constrained methods in deep architectures. And it also achieves reasonable performance compared to SOTA BN-equipped SNNs and even performs better on CIFAR10-DVS, which demonstrates its potential in processing sequential data.

---

> ### Author Response · Authors · 2025-11-24
> **Response to Reviewer s6FP [5/7]**
>
> > ***Weakness 5:*** *As depicted in Figure 2, the model incorporates both subtractive and divisive inhibition. However, the manuscript (e.g., around Equation 14) fails to provide a clear mathematical formula describing how these two operations are combined. This ambiguity severely hinders the method's reproducibility.*
>
> We have noticed the ambiguity regarding the combination of inhibitory operations. Here, we confirm that the subtractive inhibition is first performed, followed by an element-wise divisive inhibition in Equation 14.
>
> For convenience, we rewrite Equation 14 here:
>
>
> $$
> \mathbf{I}\_{\mathrm{int}}^{[l]}[t] = \mathbf{g}\_{\mathrm{E}}^{[l]} \odot \frac{\mathbf{I}\_{\mathrm{EE}}^{[l]}[t] - \mathbf{I}\_{\mathrm{EI,sub}}^{[l]}[t]}{\mathbf{I}\_{\mathrm{EI,div}}^{[l]}[t]} + \mathbf{b}\_{\mathrm{E}}^{[l]}.
> $$
>
> In Equation 14, the subtractive inhibitory current is subtracted from the excitatory input first to produce a balanced (zero-mean) input for each neuron. Then this balanced current is divided element-wise by the divisive inhibitory current to regulate the gain of the layer.
>
> **Why division after subtraction?**
>
> At first glance, it might seem strange to apply divisive inhibition to the result of the subtractive operation. In fact, this sequential order reflects the distinct locations of these two types of inhibitions occurring in a biological excitatory neuron.
>
> Biologically, subtractive inhibition typically arises from the linear summation of excitatory and inhibitory post-synaptic currents (EPSC and IPSC, respectively) at dendritic spines. Thus EPSC and IPSC are integrated at synapses, which we model as a linear subtraction [1]. Following this synaptic integration, the input current flows to the soma, where it is subjected to divisive inhibition, a process driven by increase in membrane conductance at the perisomatic region [2]. Acting as a leaky gate, this shunting effect scales the integrated currents from the dendrites before it influences the membrane potential at the soma. Therefore, the divide-after-subtract order is not arbitrary but constitutes a faithful abstraction of the current flow from dendrites to soma.
>
> To better present the process of E-I interaction, we have re-drawn Figure 2 of the manuscript, including a detailed visualization of the interactions. This revised diagram visualizes the computation where subtractive inhibition and divisive inhibition occur, and also clarifies the element-wise nature of these operations.
>
> [1] Silver, R. Neuronal arithmetic. *Nat. Rev. Neurosci.*, 2010.
>
> [2] Pouille, F. *et al.*, Enforcement of temporal fidelity in pyramidal cells by somatic feed-forward inhibition. *Science*, 2001.
>
> > ***Question 1.1:*** *Does the proposed method alter the standard forward-pass rules of SNNs?*
>
> We appreciate this question about the property of our model. We clarify that our method does not alter the standard spike-driven rules of SNNs during the forward pass. The network's feedforward backbone consists exclusively of excitatory neurons modeled as standard LIF units, ensuring that all inter-layer information transmission is totally mediated by binary spikes. While the inhibitory neurons are modelled as ReLU-like units, they function strictly as local interneurons within a layer to regulate activity and do not participate in any inter-layer connections. Therefore, our models still adhere to the spike-driven property of SNNs.

---

> ### Author Response · Authors · 2025-11-24
> **Response to Reviewer s6FP [6/7]**
>
> > ***Question 1.2:*** *Can the operations within the E-I circuit, particularly the divisive inhibition, be efficiently deployed on existing neuromorphic hardware platforms?*
>
> We appreciate this practical question. In summary, our framework is compatible with emerging  analog or mixed-signal neuromorphic hardware and would be plausible on digital chips via approximation of division like bit-shifts and look-up tables.
>
> The computation of our E-I circuit primarily consists of subtractive and divisive inhibition. The subtractive inhibition relies on inhibitory neurons projecting negative currents, which is a standard feature supported by almost all existing neuromorphic hardware (e.g., via negative synaptic weights), allowing for efficient deployment. Conversely, we acknowledge that divisive inhibition poses a challenge for current digital neuromorphic chips (such as Loihi 2), which typically lack floating-point division units. Deploying our E-I models directly on such hardware may require division approximation like bit-shifts or look-up tables.
>
> However, it is crucial to note that our divisive mechanism models shunting inhibition, a conductance-driven operation that is naturally efficient in analog or mixed-signal neuromorphic hardware. Indeed, there are hardware implementations that have successfully demonstrated the feasibility of current-mode division [1 - 3]. For instance, the DYNAP-SE2 processor [1] implements synaptic dynamics using Differential Pair Integrator (DPI) circuits. As detailed in their specifications, these circuits can perform division operations governed by the equation $\tau \dot{I}\_{out} + I_{out} = \frac{I\_{gain}}{I\_{tau}} I\_{in}$ in certain regime. This explicitly demonstrates that the divisive inhibition can be supported in such hardware.
>
> Therefore, our framework offers direct compatibility with these emerging hardware platforms. By demonstrating that local, inhibition-induced gain control can effectively replace explicit normalizations, our work validates the computational utility of these advanced hardware capabilities, motivating the design of next-generation neuron models and microarchitectures that move beyond simple integrate-and-fire dynamics.
>
> We have included this discussion in the Conclusion section of the manuscript.
>
> [1] Richter O. *et al.*, DYNAP-SE2: a scalable multi-core dynamic neuromorphic asynchronous spiking neural network processor. *Neuromorph. Comput. Eng.*, 2024.
>
> [2] Bartolozzi, C. *et al.*, Synaptic dynamics in analog VLSI. *Neural Computation*, 2007.
>
> [3] Bermak A. *et al.*, A mixed mode real-time VLSI implementation of a shunting inhibition cellular neural network, *IEEE Workshop on SiPS*, 2000.

---

> ### Author Response · Authors · 2025-11-24
> **Response to Reviewer s6FP [7/7]**
>
> > ***Question 2:*** *A core motivation of this work is the biological implausibility of BN. However, during inference, BN parameters can be folded into the weights of the preceding convolutional or fully-connected layers, meaning no standalone BN operation exists in the final network. Could you please elaborate on why BN is still considered a critical barrier to the biological plausibility of SNNs in this context?*
>
> We appreciate this question. We fully agree that BN parameters can be fused into the weights of preceding layers after training, thus removing the standalone BN operation, which mitigates the biological implausibility concern during inference.
>
> However, what we want to address is the biological implausibility of the explicit normalization operation itself, of which BN is just one specific example. Explicit normalization schemes, including BN, LN, GN, IN and so on (we mainly discuss BN in our paper since it's most frequently used in SNNs), compute the mean and standard deviation dynamically for each sample/batch. There is no evidence that biological neurons perform such explicit computations to normalize and stabilize their activity. Instead, we propose that the brain achieves a similar stabilizing effect through emergent circuit mechanisms, specifically E-I balance and gain control mediated by lateral inhibition.
>
> Therefore, the core motivation of this work extends beyond merely removing BN, and it aims to explore how cortical circuits achieve stable signal propagation under strict biological constraints. We demonstrate that by implementing the canonical E-I circuit, the network can effectively self-regulate and stabilize its activity.
>
> As for BN itself, the biological implausibility during training remains unresolved, specifically its reliance on global statistics calculated across samples in mini-batches. Furthermore, it turns out that the proposed E-I circuit and BN drive network to learn very different representations, proving the different learning dynamics of the two methods. As shown in Figure 4&7, BN forces activations into a standard Gaussian distribution, whereas our E-I circuit evolves to produce a distinct bimodal distribution driven by the separation of excitation and inhibition.
>
> In summary, while parameter fusion techniques allow for a BN-free architecture during inference, the training process remains biologically implausible. Beyond the specific case of BN, our work focuses on the broader paradigm of explicit normalization. Our primary objective is to elucidate how biological systems achieve effective signal propagation within the cortex under strict biological constraints. The fact that our network learns fundamentally different representations compared to BN-equipped networks confirms that our method captures a distinct computational regime.
>
> ---
> Above is our response to your concerns. The paper has been updated according to your comments. Thanks again for your detailed and insightful review.

---

> > ### Comment · Reviewer_s6FP · 2025-11-25
> >
> > Thank you for your excellent and detailed rebuttal. You have convincingly addressed all my concerns. The quality of the paper has been significantly improved, and I have accordingly increased my score.

---

> ### Author Response · Authors · 2025-11-25
>
> Dear Reviewer s6FP,
>
> Thank you for raising the score. Your review has helped us clarify the core motivations and deepen understandings of our work.
>
> Best regards
>
> Authors

---

### Official Review · Reviewer_p9Mj · 2025-10-27

**Soundness:** 3
**Presentation:** 3
**Contribution:** 3
**Rating:** 6
**Confidence:** 4

**Summary:**

This paper proposes a normalization-free biologically inspired method to train deep SNNs. The authors argue that most SNNs rely on non-biological normalization layers (BatchNorm, LayerNorm, etc.) to maintain activation stability, which limits biological plausibility. To overcome this, the authors propose an Excitatory-Inhibitory (E-I) architecture, including E-I init and E-I Prop, that enforces Dale’s law, meaning each neuron has a fixed excitatory or inhibitory role throughout training.

**Strengths:**

1. The aim to find a biologically plausible stabilization mechanism is well motivated. The integration of Dale’s law into deep SNNs is technically challenging and conceptually significant.
2. E-I Init and E-I Prop are technically well described and appear easy to integrate.
3. The paper could inspire further exploration into lateral inhibition and homeostatic balance as normalization-free learning tools.

**Weaknesses:**

1. Experiments are restricted to CIFAR-10/100; it remains unclear if the framework scales to large-scale or event-based datasets (e.g., ImageNet, DVS-Gesture).
2. The experiments do not include ablations isolating the impact of Dale’s-law enforcement itself. It is unclear whether improvements arise from E-I Prop (adaptive inhibition) or from Dale’s constraint.
3. The paper overclaims biological fidelity while omitting discussions of modern findings that contradict the core assumptions.
4. The training costs are not claimed. The overhead of maintaining separate excitatory and inhibitory populations and additional inhibitory weight updates is not quantified in runtime or memory terms.

**Questions:**

1. How does your method perform without enforcing Dale’s law, i.e., allowing all weights to freely take positive/negative values or other cases?
2. Could you cite recent neuroscience literature justifying Dale’s law as a valid and necessary constraint for modern models? Otherwise using a seemingly outdated theory is not sufficient for this paper.
3. What are the computational overheads (training time, memory costs, etc.) compared to normalization-based baselines? The convergence is essential for training deep SNNs.
4. Have you tested on event-based neuromorphic datasets or other dataset to validate real-world relevance?

---

> ### Author Response · Authors · 2025-11-24
> **Response to Reviewer p9Mj [1/3]**
>
> Dear Reviewer p9Mj,
>
> We are grateful for your thoughtful review and your recognition of the soundness and conceptual significance of our work. Our point-by-point responses to your questions are provided below.
>
> > ***Question 1:*** *How does your method perform without enforcing Dale’s law, i.e., allowing all weights to freely take positive/negative values or other cases?*
>
> > ***Weakness 2:*** *The experiments do not include ablations isolating the impact of Dale’s-law enforcement itself. It is unclear whether improvements arise from E-I Prop (adaptive inhibition) or from Dale’s constraint.*
>
>
> We appreciate these valuable questions. To improve ablation studies, we conducted the requested ablation study where we removed the sign constraints (allowing all weights to freely take positive or negative values) by replacing our E-I Init with standard Kaiming Initialization for $\mathbf W_{\mathrm {EE}}$ and $\mathbf W_{\mathrm {IE}}$. We found that the unconstrained models failed to converge completely under this condition. This empirical result demonstrates that E-I constraint is not optional but a functional necessity for our framework. One reason is that the strict separation of excitatory and inhibitory roles is mathematically required for our lateral inhibition mechanism to establish the stable E-I balance and gain control, which are necessary for appropriate signal propagation in deep networks.
>
> **Table R1: Ablation study with VGG-8 on CIFAR-10.**
> |E-I Constraint|E-I Init|Adap. Stab. & STE|Grad. Scaling|Top-1 Acc. (%)|
> | :--- | :--- | :--- | :--- | :--- |
> ✓|✓|✓|✓|87.03 (Ours)|
> ✓|×|✓|✓|85.61
> ✓|✓|×|✓|Failed to converge / Collapsed during training
> ✓|✓|✓|×|Failed to converge
> ×|×|✓|✓|Failed to converge
>
> We have added this new ablation study to the revised manuscript. We also re-designed and re-organized the presentation of Table 2 in the manuscript for a better presentation, as shown in Table R1. As presented in Table R1, we systematically evaluate the contribution of each component. Here, we re-summarize our ablation study to better demonstrate the effectiveness of all components of our method.
>
> **Effectiveness of E-I Init.** Replacing our initialization with standard Kaiming Init (clamped to $[0, +\infty]$) for $\mathbf W_{\mathrm {EE}}$ and $\mathbf W_{\mathrm {IE}}$ results in an accuracy drop from 87.03% to 85.61%.
>
> **Effectiveness of adaptive stabilization and STE.** Replacing our adaptive method with standard $ϵ$-stabilization results in training failure or collapse for all tested values $ϵ∈\\{10^{−8},10^{−7},10^{−6},10^{−5}\\}$.
>
> **Role of gradient scaling.** Removing the gradient scaling factor ($1/d$) for the lateral weights $\mathbf W_{\mathrm {EI}}$ immediately causes training collapse.
>
> **Necessity of E-I constraint (newly added).** Totally removing the sign constraints (allowing weights to be positive or negative) leads to complete convergence failure.

---

> ### Author Response · Authors · 2025-11-24
> **Response to Reviewer p9Mj [2/3]**
>
> > ***Question 2:*** *Could you cite recent neuroscience literature justifying Dale’s law as a valid and necessary constraint for modern models? Otherwise using a seemingly outdated theory is not sufficient for this paper.*
>
> > ***Weakness 3:*** *The paper overclaims biological fidelity while omitting discussions of modern findings that contradict the core assumptions.*
>
> We consider it an important biological and conceptual question. Here we'd like to clarify that the functional segregation of excitatory and inhibitory populations clearly exists in cortical circuits, while we have noticed that phenomena like co-transmission indeed challenge the definition of Dale's law (one neuron, one neurotransmitter) at molecular level [1]. Despite such molecular-level exceptions, physiological evidence confirms that in cortex, neurons are generally segregated into distinct excitatory (mainly Pyramidal) and inhibitory (mainly PV+, SST+, VIP+) populations [2, 3]. This segregation remains the dominant rule governing information processing in the cortex [4, 5]. Furthermore, consistent with this biological fact, many modern computational models also strictly enforce this constraint to ensure biological fidelity and facilitate analysis of E-I dynamics[6 – 10].
>
> To ensure conceptual precision, we have replaced the term "Dale's law" with terms related to E-I mechanisms in the revised manuscript. While "Dale's law" may be conceptually imprecise in the manuscript, the constraint (E-I segregation) is biologically justified for circuit modeling, and it is also what we really focus on. As demonstrated in our ablation study (Response to Question 1), the E-I segregation is not only biologically grounded but also functionally necessary for the stability of our framework.
>
> [1] Nusbaum, M. *et al*. Functional consequences of neuropeptide and small-molecule co-transmission. *Nat. Rev. Neurosci.*, 2017.
>
> [2] Markram, H. *et al*. Interneurons of the neocortical inhibitory system. *Nat. Rev. Neurosci.*, 2004.
>
> [3] Ferguson BR. *et al*. PV interneurons: critical regulators of E/I balance for prefrontal cortex-dependent behavior and psychiatric disorders. *Front. Neural Circuits*, 2018.
>
> [4] Onorato I. *et al.*. Distinct roles of PV and Sst interneurons in visually induced gamma oscillations. *Cell Rep.*, 2025.
>
> [5] Mueller-Buehl C. *et al*. Regulation of the E/I-balance by the neural matrisome. *Front. Mol. Neurosci.*, 2023.
>
> [6] Ahmadian Y. *et al.* What is the dynamical regime of cerebral cortex? *Neuron*, 2021.
>
> [7] Sadeh S. *et al.* Excitatory-inhibitory balance modulates the formation and dynamics of neuronal assemblies in cortical networks. *Sci. Adv.*, 2021.
>
> [8] Zhou S. *et al.* Synaptic E-I balance underlies efficient neural coding. *Front. Neurosci.*, 2018.
>
> [9] Tian G. *et al.* Excitation-inhibition balanced neural networks for fast signal detection. *Front. Comput. Neurosci.*, 2020.
>
> [10] Bryson A. *et al.* State transitions through inhibitory interneurons in a cortical network model. *PLOS Comput. Biol.*, 2021.

---

> ### Author Response · Authors · 2025-11-24
> **Response to Reviewer p9Mj [3/3]**
>
> > ***Question 3:*** *What are the computational overheads (training time, memory costs, etc.) compared to normalization-based baselines? The convergence is essential for training deep SNNs.*
>
> > ***Weakness 4:*** *The training costs are not claimed. The overhead of maintaining separate excitatory and inhibitory populations and additional inhibitory weight updates is not quantified in runtime or memory terms.*
>
> We appreciate this question regarding the cost of ensuring biological realism. To quantify this, we measured the training time (per epoch) and GPU memory usage on a single NVIDIA RTX 4090 for our DeepEISNN and the SNN with vanilla BN across various architectures in Table R2.
>
> **Table R2: Computational overhead comparison (T=4, Batch Size=128, Single NVIDIA RTX4090 GPU)**
> | Arch. | Metric | SNN (Baseline) | DeepEISNN (Ours) | Factor |
> | :--- | :--- | :--- | :--- | :--- |
> | VGG-8 | Time/Epoch | 7.3s | 18.1s | $2.48\times$ |
> | | Memory | 2394 MB | 2684 MB | $1.12\times$ |
> | VGG-11 | Time/Epoch | 9.5s | 23.1s | $2.43\times$ |
> | | Memory | 2392 MB | 2886 MB | $1.21\times$ |
> | VGG-16 | Time/Epoch | 16.9s | 42.2s | $2.50\times$ |
> | | Memory | 3544 MB | 5072 MB | $1.43\times$ |
> | VGG-19 | Time/Epoch | 19.4s | 47.6s | $2.45\times$ |
> | | Memory | 3734 MB | 5402 MB | $1.45\times$ |
> | ResNet-18| Time/Epoch | 38.8s | 90.0s | $2.32\times$ |
> | | Memory | 5918 MB | 9302 MB | $1.57\times$ |
>
> As shown in Table R2, our method introduces an overhead of  $2.3\times∼2.5×$ in training time and $1.1×∼1.6×$ in GPU memory usage. This overhead is an expected and necessary trade-off for ensuring biological fidelity (keep E-I ratio of 4 : 1). Unlike vanilla SNNs that use a single synaptic weight matrix per layer, our E-I circuit explicitly models three distinct synaptic projections (E -> E, E -> I, I -> E) with an additional inhibitory population. However, it is crucial to note that this overhead scales linearly with network size, making it tractable for deep architectures.
>
> As for your concern about convergence, our method demonstrates robust convergence properties despite the run-time cost, evidenced by the competitive accuracies in the updated Table 1 (please refer to the Response to All Reviewers for the full table), the E-I circuit effectively stabilizes learning process comparable to BN-equipped baselines. Therefore, this increased run-time cost grants us the ability to perform normalization-free learning using only local, biologically grounded mechanisms.
>
> According to your suggestion, we have added the runtime cost analysis above to the Appendix of the updated manuscript. Thanks for the question.
>
>
>
> > ***Question 4:*** *Have you tested on event-based neuromorphic datasets or other dataset to validate real-world relevance?*
>
> > ***Weakness 1:*** *Experiments are restricted to CIFAR-10/100; it remains unclear if the framework scales to large-scale or event-based datasets (e.g., ImageNet, DVS-Gesture).*
>
> Thanks for your suggestion to evaluate the scalability and generality of our framework beyond CIFAR10/100. Given the limited duration of the rebuttal and computational resources, training on the full ImageNet is not feasible for us. Therefore, to primarily validate the scalability of our method, we select TinyImageNet (200 classes, 64x64 input images) as a representative large-scale benchmark, along with CIFAR10-DVS and DVS-Gesture.
>
>
> As shown in updated Table 1 (please refer to the Response to All Reviewers for the full table), the ResNet-18 model achieves reasonable performance on TinyImageNet (50.29%) without fine-grained parameter tuning. Furthermore, competitive performances on event-based datasets provide further evidence of the scalability and generality of our method.
>
> ---
>
> Above is our response to your concerns. The paper has been updated according to your comments. Thanks again for your detailed and insightful review.

---

### Official Review · Reviewer_RXYj · 2025-10-31

**Soundness:** 4
**Presentation:** 2
**Contribution:** 4
**Rating:** 6
**Confidence:** 5

**Summary:**

To resolve the performance-biological realism trade-off in deep Spiking Neural Networks, this work proposes a normalization-free framework incorporating lateral inhibition.

**Strengths:**

1. It presents a novel and practical solution to a core challenge in neuromorphic computing by replacing artificial normalization with bio-plausible E-I circuits, enabling both high performance and biological fidelity.

2. The work makes substantial contributions to brain-inspired AI by rigorously implementing Dale's law and cortical microcircuit principles, providing a valuable platform for computational neuroscience exploration.

3. The framework achieves competitive accuracy on benchmark datasets while demonstrating sophisticated neural regulation through emergent bimodal activity patterns, offering a biologically grounded alternative to conventional normalization.

**Weaknesses:**

The presentation of the core methodologies, particularly the derivation of E-I Init and the custom backward pass of E-I Prop, is excessively condensed. A more structured, step-by-step exposition is needed, as the current description lacks clarity and logical progression, consequently hindering both readability and reproducibility.

**Questions:**

1. Could you please provide further explanation of the statement: "In our discrete-time simulation, the time step ∆t = 1 is chosen to be on a similar scale as the time constant of excitatory neurons (e.g., τE = 2), which implies τI ≪ ∆t. Under this condition, the dynamics of inhibitory neurons can reach a steady state almost instantaneously within a single time step"?

2. Should the weight matrices W mentioned in the text be understood as fully connected? This point requires clarification.

3. Is the variable 'g' in Equation 13 a floating-point parameter? Does the formulation involve a direct multiplication between a floating-point parameter and a floating-point weight matrix?

---

> ### Author Response · Authors · 2025-11-24
> **Response to Reviewer RXYj [1/2]**
>
> Dear Reviewer RXYj,
>
> Many thanks for your valuable feedback. We're very encourged for your recognition of soundness and contribution of our work. Below are our responses to each of the weaknesses and questions.
>
> > ***Weakness 1:*** *The presentation of the core methodologies, particularly the derivation of E-I Init and the custom backward pass of E-I Prop, is excessively condensed. A more structured, step-by-step exposition is needed, as the current description lacks clarity and logical progression, consequently hindering both readability and reproducibility.*
>
> Thank you for noting that our initial description of the core methodology may have been overly condensed due to page limits. We have revised the manuscript according to your comments, specifically adding structured, step-by-step derivations (please refer to Response to All Reviewers for the revision log). We believe these updates will greatly enhance the readability and reproducibility of our work. Below we provide detailed clarifications to your specific questions to further resolve any remaining ambiguities.
>
>
> > ***Question 1:*** *Could you please provide further explanation of the statement: "In our discrete-time simulation, the time step ∆t = 1 is chosen to be on a similar scale as the time constant of excitatory neurons (e.g., τE = 2), which implies τI ≪ ∆t. Under this condition, the dynamics of inhibitory neurons can reach a steady state almost instantaneously within a single time step"?*
>
>
> Thanks for your question. To address your concerns, we have added a rigorous, step-by-step explanation of mathematical derivation in the Appendix of the updated manuscript. Below is a step-by-step justification we have added to the revised version.
>
> Let's start from the sub-threshold dynamics of the LIF inhibitory neuron model:
>
> $$\tau_{\mathrm I}\frac{\mathrm d \mathbf u_{\mathrm I}^{[l]}(t)}{\mathrm dt}=-\left(\mathbf u_{\mathrm I}^{[l]}(t)-u_{\mathrm{I, rest}}\right)+\mathbf I_{\mathrm {I}}^{[l]}(t),$$
>
> with all notations consistent with the manuscript. By assuming
> - $u_{\mathrm{I, rest}} = 0$,
> - $\mathbf I_{\mathrm {I}}^{[l]}(t)$ is a constant $\mathbf I_0$ during the interval $[t, t+\Delta t]$,
>
> we can solve this equation and obtain
>
> $$\mathbf u_{\mathrm I}^{[l]}(t+\Delta t)=\mathbf u_{\mathrm I}^{[l]}(t)e^{-\frac{\Delta t}{\tau _\mathrm{I}}} + \mathbf I_0\left(1-e^{-\frac{\Delta t}{\tau _{\mathrm I}}}\right).$$
>
> In our simulation, excitatory neurons have $\tau_{\mathrm E}=2$, which has the same order of magnitude of $\Delta t$ ($\Delta t = 1$) in discrete-time simulation. In contrast, inhibitory neurons are modeled as fast-spiking (FS) neurons, which implies a much smaller time constant than that of excitatory neurons [1, 2], i.e., $\tau_{\mathrm I}\ll \tau_{\mathrm E} = \Theta(\Delta t)$, where $\Theta$ denotes the same order of magnitude.
>
> Under this condition, the ratio $\frac{\Delta t}{\tau _{\mathrm I}}$ becomes very large, and thus $e^{-\frac{\Delta t}{\tau _\mathrm{I}}}\rightarrow0$. By substituting the above derivation into the potential update rule, we have
>
> $$\mathbf u_{\mathrm I}^{[l]}(t+1)=\mathbf I_0,$$
>
> which means the inhibitory neurons can be effectively modeled as stateless units (i.e., "the dynamics of inhibitory neurons can reach a steady state almost instantaneously within a single time step") as long as $\tau_{\mathrm I}\ll \Delta t$, justifying our approximation in the discrete-time simulation.
>
>
> [1] Hijazi, S. *et al.*, Fast-spiking parvalbumin-positive interneurons in brain physiology and Alzheimer’s disease. *Mol. Psychiatry*, 2023.
>
> [2] Milicevic, KD. *et al.*, Physiological features of parvalbumin-expressing GABAergic interneurons contributing to high-frequency oscillations in the cerebral cortex. *Current Research in Neurobiology*, 2024.

---

> ### Author Response · Authors · 2025-11-24
> **Response to Reviewer RXYj [2/2]**
>
> > ***Question 2:*** *Should the weight matrices W mentioned in the text be understood as fully connected? This point requires clarification.*
>
> We apologize for the ambiguity regarding the definition of $\mathbf{W}$. For the sake of brevity, the mathematical formulation in our original manuscript primarily used the notation of a fully connected architecture, where all synaptic weights ($\mathbf W_\mathrm{EE}, \mathbf W_\mathrm{IE}, \mathbf W_\mathrm{EI}$) are represented as 2-D matrices.
>
> In experiments, we mainly use CNNs like VGG and ResNet, where all convolution layers and fully connected classifier are implemented with the proposed E-I circuit. Here we would like to clarify the implementation of the convolution layer used in our experiments. Specifically, $\mathbf W_\mathrm{EE}$ and $\mathbf W_\mathrm{IE}$ are standard $K\times K$ convolution kernels, while the lateral connection $\mathbf W_\mathrm{EI}$ is a $1\times 1$ point-wise convolution kernel. In this context, the input dimension $d = C_{\mathrm{in}}\times K \times K$ where $C_{\mathrm{in}}$ is #channels of excitatory input, while the population sizes $n_\mathrm E$ and $n_\mathrm I$ are the number of excitatory and inhibitory output channels, respectively. This configuration allows inhibitory neurons to regulate excitatory neurons densely across the channel dimension while preserving the spatial structure.
>
> We have updated the manuscript to clarify that our mathematical formulation is based on a fully connected architecture for brevity. Additionally, we have included the detailed implementation of E-I circuits for convolution layers in the Appendix.
>
> > ***Question 3:*** *Is the variable 'g' in Equation 13 a floating-point parameter? Does the formulation involve a direct multiplication between a floating-point parameter and a floating-point weight matrix?*
>
> To clarify, $\mathbf g_{\mathrm I}^{[l]}$ is not a scalar but a trainable floating-point vector with dimensionality of $n_{\mathrm I}^{[l]}$, i.e., $\mathbf g_{\mathrm I}^{[l]} \in \mathbb R^{n_{\mathrm I}^{[l]}}$. And it is also kept non-negative during training to ensure inhibitory outputs.
>
> As for the Eq. 13, the operation does not involve a direct multiplication between the parameter $\mathbf g_{\mathrm I}^{[l]}$ and weight matrix $\mathbf W_\mathrm{EI}^{[l]}\in\mathbb R^{n_{\mathrm E}^{[l]}\times n_{\mathrm I}^{[l]}}$. Instead, the first computation is performing an element-wise multiplication (Hadamard product, denoted by $\odot$ in the paper) between $\mathbf g_{\mathrm I}^{[l]}$ and inhibitory signal $\mathbf s_{\mathrm I}^{[l]}[t]$, representing a gated inhibitory activity. Then this product is multiplied by $\mathbf W_\mathrm{EI}^{[l]}$ to compute the divisive inhibitory currents.
>
> We have clarified the dimensionality of $\mathbf W_\mathrm{EI}^{[l]}$ and $\mathbf g_{\mathrm I}^{[l]}$ in the revised manuscript. Thanks for pointing out our oversight.
>
> ---
>
> Above is our response to your concerns. The paper has been updated according to your comments. Thanks again for your detailed and insightful review.

---

### Official Review · Reviewer_L1V9 · 2025-11-02

**Soundness:** 4
**Presentation:** 4
**Contribution:** 3
**Rating:** 8
**Confidence:** 5

**Summary:**

This paper revisits the important role of lateral inhibition in the cortical circuit and develops a normalization-free learning framework. The authors also developed a useful dynamic initialization scheme to ensure effective learning. The proposed framework supports the effective training of deep networks and shows competitive results on multiple benchmarks. Overall, I find this paper is quite interesting and the main idea is easy to follow.

**Strengths:**

- The work addresses the critical challenge of training deep SNNs without biologically implausible normalization schemes like Batch Normalization. The proposed framework, inspired by lateral inhibition in cortical circuits, is a significant step towards bridging the gap between high-performance machine learning and biological realism.

- The paper is well-written, and the core concepts are presented in a way that is easy to understand. The motivation for replacing standard normalization with a brain-inspired alternative is clearly articulated.

**Weaknesses:**

- While the performance has been validated on static image classification tasks, I think it would be better to further examine the performance on more extensive neuromorphic datasets for a broader impact.
- It would be great to discuss the potential deployment on existing neuromorphic hardware.
- Table 1 lacks statistical significance analysis.

**Questions:**

The authors incorporate Dale’s law into their models. I fully understand their consideration from a biological perspective. Beyond this, I wonder if the authors observed any other functional implications or accuracy gains by following this biological law?

---

> ### Author Response · Authors · 2025-11-24
> **Response to Reviewer L1V9 [1/2]**
>
> Dear Reviewer L1V9,
>
> Thank you for your detailed and insightful comments. We're encouraged by your positive comment that our paper is interesting and our method is easy to follow. Below is our response to each of your concerns.
>
>
> > ***Weakness 1:*** *While the performance has been validated on static image classification tasks, I think it would be better to further examine the performance on more extensive neuromorphic datasets for a broader impact.*
>
> We appreciate your suggestion to evaluate our framework's generality on neuromorphic datasets. To demonstrate the effectiveness of our method on event-driven tasks, we conduct experiments on DVS-Gesture and CIFAR10-DVS. We utilize the exactly same configuration as in CIFAR-10 experiments, with the only exception of adjusting the learning rate from 0.002 to 0.001.
>
> As shown in the updated Table 1 (please refer to the Response to All Reviewers for the full table), models trained with our method successfully converge on both datasets, without fine-grained parameter selections. These results provide evidence that our E-I framework is not only limited to static images but is also effective and robust for processing event streams.
>
>
> > ***Weakness 2:*** *It would be great to discuss the potential deployment on existing neuromorphic hardware.*
>
> We appreciate this practical question. In summary, our framework is compatible with emerging  analog or mixed-signal neuromorphic hardware and would be plausible on digital chips via approximation of division like bit-shifts and look-up tables.
>
> Computation of the E-I circuit mainly consists of subtractive and divisive inhibition. The former relies on inhibitory neurons projecting negative currents, which is supported by almost all existing neuromorphic hardware (e.g., through negative synaptic weights).
>
> For the divisive inhibition, we acknowledge that it is a challenge for current digital neuromorphic chips (such as Loihi 2), which typically lack large-scale floating-point division units. Deploying our E-I models directly on such digital chips would require approximation like bit shifts and look-up tables. On the other hand, there are analog or mixed-signal hardware implementations that demonstrate the feasibility of current-mode division operations [1 - 3]. For example, the DYNAP-SE2 [1] implements synaptic dynamics using Differential Pair Integrator (DPI) circuits, which can perform scaling operations with the form $\tau \dot{I}\_{out} + I\_{out} = \frac{I\_{gain}}{I\_{tau}} I\_{in}$. This explicitly demonstrates that divisive scaling can be implemented in hardware.
>
> Therefore, our framework offers compatibility with these emerging hardware. By demonstrating that local inhibition-induced gain control can effectively replace explicit normalizations, our work validates the computational utility of these advanced hardware capabilities, motivating the design of next-generation neuron models and microarchitectures that move beyond simple integrate-and-fire dynamics.
>
> We have included this discussion in the Conclusion section of the manuscript. Thank you again for raising such a valuable question.
>
> [1] Richter O. *et al.*, DYNAP-SE2: a scalable multi-core dynamic neuromorphic asynchronous spiking neural network processor. *Neuromorph. Comput. Eng.*, 2024.
>
> [2] Bartolozzi, C. *et al.*, Synaptic dynamics in analog VLSI. *Neural Computation*, 2007.
>
> [3] Bermak A. *et al.*, A mixed mode real-time VLSI implementation of a shunting inhibition cellular neural network, *IEEE Workshop on SiPS*, 2000.

---

> ### Author Response · Authors · 2025-11-24
> **Response to Reviewer L1V9 [2/2]**
>
> > ***Weakness 3:*** *Table 1 lacks statistical significance analysis.*
>
> We agree that statistical analysis is crucial for evaluating the robustness of our method. To address this, we have conducted $N=5$ independent runs using different random seeds for all experiments in Table 1 of the original manuscript, including evaluation of SNN with vanilla BN, DANN, and our DeepEISNN.
>
> **Table R1: Performance statistics (Mean ± Std over 5 runs) on CIFAR-10/100.***
> |Dataset| Model |Type|E-I | Norm. |Arch. | T | Top-1 Acc.(%) |
> | :--- |:--- |:--- | :--- | :--- | :--- | :--- | :--- |
> |CIFAR10|DANN|ANN|Y|N|VGG8|$-$|86.01 ± 0.32|
> ||||||VGG11|$-$|86.05 ± 0.16|
> ||||||VGG16|$-$|88.54 ± 0.38|
> ||||||VGG19|$-$|88.28 ± 0.27|
> ||SNN with vanilla BN|SNN|N|Y|VGG16|4|94.29 ± 0.07|
> ||||||VGG19|4|94.11 ± 0.15|
> ||||||ResNet18|4|95.37 ± 0.13|
> ||**DeepEISNN (Ours)**|SNN|Y|N|VGG8|4|**86.88 ± 0.32**|
> ||||||VGG11|4|**88.26 ± 0.25**|
> ||||||VGG16|4|**90.80 ± 0.16**|
> ||||||VGG19|4|**90.93 ± 0.33**|
> ||||||ResNet18|4|**92.05 ± 0.11**|
> |CIFAR100|**DeepEISNN (Ours)**|SNN|Y|N|VGG16|4|**64.95 ± 1.14**|
> ||||||VGG19|4|**64.31 ± 0.41**|
>
> As illustrated in Table R1, our DeepEISNN consistently outperforms the DANN baseline across all VGG architectures on CIFAR10. To verify the statistical significance of these improvements, we performed an independent two-sample Welch’s t-test between our method and DANN for each architecture. Results are shown in Table R2.
>
> **Table R2: t-test results comparing DeepEISNN (Ours) vs. DANN on CIFAR-10.**
> |Arch.    | t-statistic  | p-value      | Significance|
> | :-------------| :-------------| :------------|:--------------|
> VGG-8           |     4.2987   |   2.62e-03   | ** (p < 0.01)
> VGG-11          |    16.6490   |   9.11e-07   | *** (p < 0.001)
> VGG-16          |    12.2566   |   3.93e-05   | *** (p < 0.001)
> VGG-19          |    13.8974   |   9.99e-07   | *** (p < 0.001)
>
> As shown in Table R2, p-values for all architectures are well below the standard significance level (α=0.05). This validates our E-I constrained framework achieves a statistically significant performance gain over DANN.
>
> We have included these statistical analyses in the Appendix of the revised manuscript. Thanks again for your suggestion.
>
> > ***Question 1:*** *The authors incorporate Dale’s law into their models. I fully understand their consideration from a biological perspective. Beyond this, I wonder if the authors observed any other functional implications or accuracy gains by following this biological law?*
>
>
>
> Thank you for the question. A key **functional implication** is that the inhibitory signal actively forces the network to learn sparse representations. As illustrated in Figure 4 of the manuscript, the E-I circuit evolves and produces a bimodal input distribution after training. This bimodal distribution can be interpreted as a mixture of two Gaussian-like components. Notably, there is always a component centered in the negative regime, which is driven by inhibitory signals, effectively pushing irrelevant neurons into a sub-threshold (silent) state. This selective suppression mechanism directly leads to the sparse and edge-centric feature maps shown in Figure 8 of the manuscript.
>
> As for **accuracy**, Dale’s law contributes to network stability and scalable accuracy. Although constraining weights to a fixed sign narrows the optimization space, it also establishes the structural requirement for a balanced excitatory-inhibitory regime. This balance allows our E-I Init to precisely match and cancel currents, preventing runaway excitation or vanishing activity, which may effectively prevent the exploding/vanishing signals in deep SNNs and thereby allow us to scale up to deeper architectures (e.g., ResNet-18) and ultimately achieve higher accuracy than shallow, unconstrained baselines.
>
> ---
> Above is our response to your concerns. The paper has been updated according to your comments. Thanks again for your detailed and insightful review.

---

> ### Comment · Reviewer_L1V9 · 2025-11-27
>
> I really appreciate the author's great efforts and detailed responses to my questions, which have convincingly addressed all my concerns.

---

> > ### Author Response · Authors · 2025-11-27
> >
> > Thank you once again for your recognition of our work. Your review has helped us make the experiment results more convincing and our work more comprehensive.

---

### Author Response · Authors · 2025-11-24
**Response to All Reviewers [1/2]: Evaluation on Neuromorphic/Large-Scale Datasets and Comparisons with SOTA Methods**

Dear Reviewers,

We sincerely thank all reviewers for the time and efforts to provide detailed and insightful comments on our work.

A common suggestion by most reviewers is to evaluate the generality of our framework on neuromorphic datasets (by Reviewer L1V9, p9Mj, s6FP) and large-scale datasets (by Reviewer p9Mj), and to provide a broader comparison with SOTA methods (by Reviewer s6FP). To address these concerns, we have conducted additional experiments on CIFAR10-DVS, DVS-Gesture, and added more comparison with SOTA SNNs with normalization. To further investigate our method's scalability, we selected TinyImageNet as a representative large-scale benchmark, given the limited time of the rebuttal period. Here, we summarize these new results, which have been integrated into Table 1 of the revised manuscript. As shown below, our methods can effectively scale up to event-based and large-scale datasets, demonstrating the generality of the proposed framework.

**Table 1 (Updated): Comparison with E-I constrained, normalization-free, and BN-equipped SNNs.**

|Dataset|E-I | Norm. | Method |Type|Arch. | T | Top-1 Acc.(%) |
|:--- |:--- |:--- | :--- | :--- | :--- | :--- | :--- |
|CIFAR-10|×|×|DAP (Micheli et al., 2025)|SNN|7-layer CNN|3|57.52|
||||SRI (Ding et al., 2025)|SNN|VGG-9|20|87.62|
||||B-SNN (Karimah et al., 2025) |SNN|VGG-8|64|87.73|
||||EICIL (Shao et al., 2023)|SNN|ResNet-18|$-$|90.34|
||||IM-Loss (Guo et al., 2022)| SNN |CIFARNet| 4| 90.90|
||×|✓|*Vanilla BN|SNN|VGG-16| 4|94.29 ± 0.07|
||||||VGG-19| 4 |94.11 ± 0.15|
||||||ResNet-18| 4 |95.37 ± 0.13|
||||BNTT (Kim & Panda, 2021)|SNN|VGG-9| 20 |90.30|
||||NeuNorm (Wu et al., 2019)|SNN|CIFARNet|12|90.53|
||||TEBN (Duan et al., 2022)|SNN|VGG-9|4|92.81|
||||||ResNet-19|4|94.70|
||||tdBN (Zheng et al., 2021) |SNN|ResNet-19 |4|92.92|
||||TAB (Jiang et al., 2024)|SNN|VGG-9|4|93.41|
||||||ResNet-19|4|94.76|
||✓|×|FDI (Rossbroich et al., 2022)|SNN|6-layer CNN| 50| 65.60|
||||*DANN (Cornford et al., 2021)|ANN|VGG-16|$-$|88.54 ± 0.38|
||||||VGG-19|$-$|88.28 ± 0.27|
||||EICIL (Shao et al., 2023)|SNN|E-I Net|$-$|89.43|
||||BackEISNN (Zhao et al., 2022)|SNN|5-layer CNN|20|90.93|
||||**DeepEISNN (Ours)**|SNN|VGG-16|4|**90.80 ± 0.16**|
||||||VGG-19|4|**90.93 ± 0.33**|
||||||ResNet-18|4|**92.05 ± 0.11**|
|CIFAR-100|×|×|SRI (Ding et al., 2025)|SNN|VGG-11|20|54.94|
||||EICIL (Shao et al., 2023)|SNN|ResNet-18|$−$|63.47|
||×|✓|BNTT (Kim & Panda, 2021)|SNN|VGG-11|50|66.60|
||||TEBN (Duan et al., 2022)|SNN|VGG-11|4|74.37|
||||TAB (Jiang et al., 2024)|SNN|VGG-11|4|75.89|
||✓|×|EICIL (Shao et al., 2023)|SNN|E-I Net|$−$|53.86|
||||**DeepEISNN (Ours)**|SNN|VGG-16|4|**64.95 ± 1.14**|
||||||VGG-19|4|**64.31 ± 0.41**|
|CIFAR10-DVS|×|×|IM-Loss (Guo et al., 2022)|SNN|ResNet-19|10|72.60
||×|✓|NeuNorm (Wu et al., 2019)|SNN|7-layer CNN|40|60.50|
||||BNTT (Kim & Panda, 2021)|SNN|7-layer CNN|20|63.20|
||||tdBN (Zheng et al., 2021)|SNN|ResNet-19|10|67.80|
||||TEBN (Duan et al., 2022)|SNN|7-layer CNN|10|75.10|
||||TAB (Jiang et al., 2024)|SNN|7-layer CNN|4|76.70|
||✓|×|**DeepEISNN (Ours)**|SNN|VGG-8|10|**77.30 ± 0.64**|
||||||VGG-11|10|**77.66 ± 0.48**|
|DVS-Gesture|✓|×|FDI (Rossbroich et al., 2022)|SNN|6-layer CNN|500|86.7|
||||**DeepEISNN (Ours)**|SNN|VGG-8|16|**94.86 ± 0.86**|
|TinyImageNet|✓|×|**DeepEISNN (Ours)**|SNN|ResNet-18|4|**50.29**|

\* Methods marked with asterisk are reproduced by us (averaged over 5 runs). Other results are cited from existing literature.

**Highlights of the Update**
- **Evaluation on neuromorphic datasets.** We demonstrate that our method achieves competitive performance on CIFAR10-DVS and DVS-Gesture, confirming the generalization capability of our framework. Surprisingly, our model outperforms SOTA SNNs with normalization on CIFAR10-DVS, which suggests the inherent robustness of our E-I circuit in processing spatiotemporal dynamics.
- **Evaluation on large-scale datasets.** Suggested by Reviewer p9Mj, we evaluate our method on TinyImageNet. Our ResNet-18 model achieved a Top-1 accuracy of 50.29%, demonstrating the effectiveness of our methods on larger datasets.
- **Performance statistics.** Suggested by Reviewer L1V9, we now report the mean ± standard deviation over 5 independent runs for our method (DeepEISNN) and key baselines (DANN, Vanilla BN).
- **Comparisons with SOTA SNNs.** Suggested by Reviewer s6FP, we have included comparisons with recent BN-equipped SOTA methods (e.g., TEBN and TAB) to provide a comprehensive performance comparison.

We hope these additional results will address your concerns regarding the evaluation and performance. Specific responses to each reviewer's questions can be found in the corresponding replies. We remain available to answer any further questions.

---

### Author Response · Authors · 2025-11-24
**Response to All Reviewers [2/2]: Paper Revision Log**

Dear Reviewers,

We have uploaded a revised version of our manuscript to OpenReview. To facilitate your review, all major changes incorporating your suggestions have been highlighted in blue. For your convenience, we summarize these key revisions and new contents in each section below.

**Abastract & Introduction**

- [p9Mj] Replaced the term "Dale's law" with terminology focusing on "E-I segregation" to improve conceptual precision and better elucidate the focus (E-I interactions) of our work.
- [s6FP] Broadened the scope to emphasize general normalization schemes rather than focusing exclusively on BN.

**Related Work**
- [s6FP] Expanded the discussion to include additional literature on SNNs utilizing normalization techniques.

**Preliminary**
- [p9Mj] Revised to introduce E-I segregation in cortical circuits instead of Dale's law, ensuring conceptual accuracy.

**Methods**
- [RXYj, s6FP] Redesigned Figure 2 to clearly illustrate the computational process of the E-I circuit and to better align with Eq 14.
- [RXYj] Added a specific clarification regarding the architecture discussed in the main text.
- [RXYj] Included a new diagram (Figure 3) to improve the readability and understanding of the E-I Prop mechanism.

**Experiments**
- [L1V9, p9Mj, s6FP] Updated experimental results (see Response to All Reviewers [1/2]) and reformatted Table 1 for improved presentation.
- [p9Mj] Included an ablation study validating the necessity of the E-I constraint and reorganized Table 2.
- [L1V9] Elaborated on the discussion regarding the emergent bimodal distribution as a functional implication of E-I circuits.

**Conclusion and Discussion**
- [p9Mj] Refined terminology by replacing "Dale's law" with E-I related terms to maintain conceptual precision.
- [s6FP] Reaffirmed our focus on general normalization schemes.
- [L1V9, s6FP] Added a discussion regarding the potential deployment of our framework on neuromorphic hardware.

**Appendix**
- [RXYj, s6FP] Added a mathematical proof demonstrating the rigor of approximating fast-spiking inhibitory neurons as ReLU-like units.
- [RXYj] Provided step-by-step derivations for E-I Init to enhance clarity and presentation.
- [s6FP] Included detailed backpropagation derivations to support the analysis of gradient scaling.
- [RXYj] Added implementation details regarding the application of E-I circuits in CNNs.
- [RXYj] Included the algorithm/pseudocode for E-I Init to facilitate - implementation.
- [s6FP] Added visualizations of gradient magnitudes alongside both theoretical and empirical analyses of gradient scaling.
- [L1V9] Included statistical significance analysis results for the comparison with DANN.
- [p9Mj] Added analysis of runtime overhead.
- Provided supplementary experimental details to ensure reproducibility.

In addition to these major updates, we have also addressed minor issues, including typo corrections, writing improvements, and figure refinements for better clarity.

We sincerely appreciate your time and patience throughout this process.

---

### Author Response · Authors · 2025-12-01
**To PC, SAC, and AC: Discussion Summary**

Dear PC, SAC, AC,

We sincerely appreciate the great efforts and time you spend on checking each submission. Here, we would like to provide a brief summary on the rebuttal status to assist your decision-making.

|Reviewer|Initial Rating|Rating before Leak|Status|Note|
|:---:|:---:|:---:|:--|:--|
|`L1V9`|8 [4, 4, 3]|8 [4, 4, 3]|Replied [Nov 27]|All concerns resolved, rating maintained|
|`RXYj`|6 [4, 2, 4]|6 [4, 2, 4]|**No Reply**| All concerns answered|
|`p9Mj`|6 [3, 3, 3]|6 [3, 3, 3]|**No Reply**| All concerns answered|
|`s6FP`|4 [3, 2, 2]|**6 [3, 4, 3]**|Replied [Nov 25]|All concerns resolved, rating increased [4 -> 6]|
|Average|**6**|**6.5**|$-$|$-$|

- *All replies and rating updates occurred before the information leak incident on Nov 27.*
- *[S, P, C] denotes [Soundness, Presentation, Contribution].*
---
**Our requests**

- Given the lack of response from `RXYj` and `p9Mj`, we would appreciate it if you could review our rebuttals, as we are confident that their concerns have been fully resolved.
- We kindly invite you to verify the timestamp of Reviewer `s6FP`'s rating increase in the discussion.

For your convenience, we have provided some details about our work and the discussion with reviewers below.

---
**Core contributions of our work**
- **Novel Framework:** We introduce a biologically grounded framework that incorporates cortical E-I circuits into deep SNNs. It enables effective normalization-free training, providing an ideal platform for further exploring large-scale cortical computation.
- **Technical Contribution:** We develop E-I Init and E-I Prop to enable effective training of deep SNNs without normalizations, which has long been a critical challenge in SNN literature. These techniques prove to be necessary for effective training.
- **Superior Performance**: To the best of our knowledge, our method achieves SOTA performance among normalization-free SNNs. Crucially, it demonstrates high performance on neuromorphic datasets, outperforming BN-equipped SOTA methods on CIFAR10-DVS.

**Our response to main concerns of each reviewer**

- Reviewer `L1V9`
    - **Validation on Neuromorphic Datasets:** We conducted additional experiments on DVS-Gesture and CIFAR10-DVS to demonstrate the framework's effectiveness on event-driven tasks.
    - **Hardware Deployment:** We added a discussion clarifying that our framework is compatible with analog/mixed-signal hardware and feasible on digital chips via approximation (e.g., bit-shifts and lookup tables).
    - **Statistical Significance:** We performed 5 independent runs and t-tests, confirming that our improvements over baselines are statistically significant (p<0.05).
    - **Functional Implication:** Unlike the Gaussian distribution enforced by Batch Normalization, our framework evolves to exhibit a bimodal distribution. This emergent property confirms that E-I interactions actively perform gain control through a mechanism distinct from explicit normalization schemes.
- Reviewer `RXYj`
    - **Methodology Clarity:** We added step-by-step derivations and pseudocode in the Appendix to address concerns about the condensed presentation.
    - **Validity of Approximation:** We provided a mathematical proof justifying that inhibitory neurons effectively behave as stateless units when $τ_\mathrm I≪ Δt$, validating our discrete-time approximation.
    - **Implementation Details:** We clarified that lateral connections in CNNs use 1×1 pointwise convolution and defined the specific dimensionality of trainable parameters.
- Reviewer `p9Mj`
    - **Necessity of E-I Constraint:** We added an ablation study showing that removing sign constraints (allowing weights to freely take positive/negative values) leads to convergence failure, proving the constraint is functionally necessary.
    - **Conceptual Precision:** We changed the term "Dale's law" to terms better focusing on E-I interactions.
    - **Runtime Overhead:** We quantified the training overhead, demonstrating that it is a tractable trade-off for biological fidelity.
    - **Scalability:** We extended evaluations to TinyImageNet and neuromorphic datasets, demonstrating the method's generality.
- Reviewer `s6FP`
    - **Validity of Approximation:** We justified the ReLU-like inhibitory model via mathematical derivation (no error accumulation) and empirical verification (performance holds from T=1 to 16).
    - **Gradient Scaling:** We provided both theoretical and empirical analysis showing the 1/d scaling is necessary to counter gradient amplification in the divisive term.
    - **SOTA Comparison:** We added comparisons with BN-equipped SOTA SNNs (e.g., TEBN, TAB), showing our method is competitive on static images and superior on CIFAR10-DVS.
    - **Clarification on E-I Operations:** We clarified that subtractive inhibition precedes divisive inhibition and redrew Figure 2 to improve clarity.
---
Thank you again for your dedication to safeguarding the fairness of the review process.

Warm regards,

Authors

---

### Meta-Review · Area_Chair_n77S · 2025-12-22

**Summary:**

The authors propose DeepEISNN, a biologically grounded framework designed to train deep Spiking Neural Networks (SNNs) without relying on biologically implausible normalization layers like Batch Normalization (BN). To achieve this, they introduce a lateral inhibition mechanism inspired by cortical Excitatory-Inhibitory (E-I) circuits, consisting of distinct excitatory and inhibitory populations. The framework is supported by two key technical contributions: E-I Init, a dynamic initialization scheme to balance inputs, and E-I Prop, a backpropagation method that regulates gradient flow through the E-I circuits. The goal is to bridge the gap between high-performance deep learning and biological realism by enabling stable end-to-end training of deep SNNs without explicit normalization.

The paper was well-received for its motivation and novel approach to replacing artificial normalization with biological mechanisms. Initial reviewer concerns focused on the scalability of the method to larger or event-based datasets, the validity of simplifying inhibitory neurons to stateless ReLU-like units, the necessity of the E-I constraints, and a lack of statistical significance analysis. In response, the authors provided a comprehensive rebuttal that included new experiments on CIFAR10-DVS, DVS-Gesture, and TinyImageNet, alongside mathematical proofs justifying their approximations and ablation studies confirming that E-I constraints are functionally necessary for convergence.

**Reviewer Concerns:**

### **Addressed Concerns**

**Scalability and Datasets**: Reviewers L1V9, p9Mj, and s6FP requested evaluation on neuromorphic and larger datasets. The authors added results for CIFAR10-DVS, DVS-Gesture, and TinyImageNet. The method achieved competitive performance, outperforming some BN-equipped SOTA models on CIFAR10-DVS.

**Statistical Significance**: Reviewer L1V9 noted a lack of statistical analysis. The authors performed 5 independent runs and provided t-test results demonstrating statistically significant improvements ($p < 0.05$) over baselines like DANN.

**Validity of Approximations**: Reviewer s6FP and RXYj questioned the "stateless ReLU" approximation for inhibitory neurons and the "1/d" gradient scaling. The authors provided a mathematical proof showing that when $\tau_I \ll \Delta t$, inhibitory dynamics reach steady state instantaneously, justifying the approximation. They also provided empirical data showing performance holds for small time steps ($T=1$).

**Necessity of Constraints**: Reviewer p9Mj asked if the E-I constraints (Dale's Law) were necessary. The authors added an ablation study showing that removing sign constraints (allowing free weights) causes the model to fail to converge, proving the constraints are functionally necessary.

**Comparison with SOTA**: Reviewer s6FP requested comparisons with BN-equipped SOTA SNNs. The authors updated Table 1 to include methods like TEBN and TAB, showing DeepEISNN is competitive on static tasks and superior on DVS tasks.

### **Remaining Concerns**

**Clean Accuracy Gap**: While the authors showed competitive results, there is still a slight performance gap (approx. 2-3%) compared to the absolute best BN-equipped baselines on static images. The authors argue this is acceptable given the biological constraints, but it remains a numerical reality.

**Hardware Deployment**: While the authors discussed compatibility with analog hardware, digital deployment requires approximations (e.g., bit-shifts) for the divisive inhibition, which might not be trivial to implement efficiently on all current platforms.

**Reviewer Scores:**

**Reviewer L1V9: 8 (Accept).**

Rationale: This reviewer was already positive (initial 8) and confirmed that "detailed responses... have convincingly addressed all my concerns". The new statistical analysis and neuromorphic experiments solidified this high score.

**Reviewer s6FP: 4 $\rightarrow$ 5/6 (Borderline or Accept).**
Rationale: Originally a 4, this reviewer explicitly stated, "I have accordingly increased my score" to 6 after the rebuttal. They were satisfied with the SOTA comparisons, approximations justification, and clarifications on the inhibition mechanism.

**Reviewer RXYj: 6 (Accept).**
Rationale: The reviewer found the initial presentation condensed. The authors added detailed derivations and pseudocode to the appendix. Since the reviewer's concerns were primarily about clarity and presentation—which were addressed—the score is likely to hold or improve.

**Reviewer p9Mj: 6 (Accept).**
Rationale: This reviewer questioned the necessity of constraints and requested cost analysis. The ablation study proving convergence failure without E-I constraints and the detailed cost table directly answered these points. The score is solid at 6.

---

### Decision · Program_Chairs · 2026-01-26

Accept (Poster)